# Multiple New Strains of Amphidomataceae (Dinophyceae) from the North Atlantic Revealed a High Toxin Profile Variability of *Azadinium spinosum* and a New Non-Toxigenic *Az.* cf. *spinosum*

**DOI:** 10.3390/microorganisms9010134

**Published:** 2021-01-08

**Authors:** Urban Tillmann, Stephan Wietkamp, Haifeng Gu, Bernd Krock, Rafael Salas, Dave Clarke

**Affiliations:** 1Helmholtz Center for Polar and Marine Research, Alfred Wegener Institute, Am Handelshafen 12, D-27570 Bremerhaven, Germany; stephan.wietkamp@awi.de (S.W.); Bernd.Krock@awi.de (B.K.); 2Third Institute of Oceanography, Ministry of Natural Resources, Xiamen 361005, China; guhaifeng@tio.org.cn; 3School of Marine Sciences, Nanjing University of Information Science and Technology, Nanjing 210044, China; 4Marine Institute, Rinville, Oranmore, H91 R673 Co. Galway, Ireland; Rafael.Salas@Marine.ie (R.S.); Dave.Clarke@Marine.ie (D.C.)

**Keywords:** azaspiracids, toxin profile, toxin cell quota, variability, ribotype

## Abstract

Azaspiracids (AZA) are a group of lipophilic toxins, which are produced by a few species of the marine nanoplanktonic dinoflagellates *Azadinium* and *Amphidoma* (Amphidomataceae). A survey was conducted in 2018 to increase knowledge on the diversity and distribution of amphidomatacean species and their toxins in Irish and North Sea waters (North Atlantic). We here present a detailed morphological, phylogenetic, and toxinological characterization of 82 new strains representing the potential AZA producers *Azadinium spinosum* and *Amphidoma languida*. A total of ten new strains of *Am. languida* were obtained from the North Sea, and all conformed in terms of morphology and toxin profile (AZA-38 and-39) with previous records from the area. Within 72 strains assigned to *Az. spinosum* there were strains of two distinct ribotypes (A and B) which consistently differed in their toxin profile (dominated by AZA-1 and -2 in ribotype A, and by AZA-11 and -51 in ribotype B strains). Five strains conformed in morphology with *Az. spinosum,* but no AZA could be detected in these strains. Moreover, they revealed significant nucleotide differences compared to known *Az. spinosum* sequences and clustered apart from all other *Az. spinosum* strains within the phylogenetic tree, and therefore were provisionally designated as *Az.* cf. *spinosum*. These *Az*. cf. *spinosum* strains without detectable AZA were shown not to cause amplification in the species-specific qPCR assay developed to detect and quantify *Az. spinosum*. As shown here for the first time, AZA profiles differed between strains of *Az. spinosum* ribotype A in the presence/absence of AZA-1, AZA-2, and/or AZA-33, with the majority of strains having all three AZA congeners, and others having only AZA-1, AZA-1 and AZA-2, or AZA-1 and AZA-33. In contrast, no AZA profile variability was observed in ribotype B strains. Multiple AZA analyses of a period of up to 18 months showed that toxin profiles (including absence of AZA for *Az.* cf. *spinosum* strains) were consistent and stable over time. Total AZA cell quotas were highly variable both among and within strains, with quotas ranging from 0.1 to 63 fg AZA cell^−1^. Cell quota variability of single AZA compounds for *Az. spinosum* strains could be as high as 330-fold, but the underlying causes for the extraordinary large variability of AZA cell quota is poorly understood.

## 1. Introduction

Azaspiracids (AZA) are polyether lipophilic marine biotoxins that accumulate in filter-feeding bivalves. AZA have been associated with human incidents of shellfish poisoning since the first intoxication case in 1995 attributed to Irish mussels [1,2]. To date, seven human azaspiracid shellfish poisoning (AZP) events have been confirmed in The Netherlands, Ireland, Italy, France, the UK, and the US [3,4], and each of these AZP events have been traced to contaminated Irish shellfish.

Although AZA have now been reported in shellfish and/or plankton samples from numerous geographical sites of the Atlantic [5,6,7,8,9,10] and Pacific [11,12,13,14,15], Ireland, with its important shellfish industry, remains the country most seriously affected by AZA related problems. AZA concentrations above the EU threshold level of 0.16 mg kg^−1^ shellfish meat have exceptionally been recorded in Norway in 2002/2003 [16] and along the Atlantic coast of southern Spain in 2009 [17]. However, elevated AZA levels in Ireland have led to recurrent and extended production site closures with severe economic consequences for the shellfish industry since 2002 [18,19,20]. Accordingly, there is a need to increase knowledge on the diversity and distribution of AZA-producing species in order to allow better identification and quantification of these species in national monitoring programs. Moreover, detailed knowledge of the biology, toxin profile, cell quota, and the regulating factors of local populations is required for a better understanding of bloom formation, bloom dynamics, and bloom impacts on shellfish and the marine environment.

With the formal description of the new small dinoflagellate *Azadinium spinosum* Elbrächter and Tillmann (Amphidomataceae), the first source organism of AZA was identified [21]. Amphidomataceae include *Azadinium* and the closely related genus *Amphidoma*, and now more than 30 amphidomatacean species are known, of which only four, *Az. spinosum*, *Az. poporum* Tillmann and Elbrächter, *Az. dexteroporum* Percopo and Zingone, and *Amphidoma languida* Tillmann, Salas and Elbrächter, are known to produce AZA [21,22,23]. The first strains of *Az. spinosum* originating from Scotland, Denmark, and the Shetland Islands all share the same toxin profile, i.e., AZA-1, AZA-2, and AZA-33 [24,25], as did the first and only strain isolated from Irish waters in 2011 [18].

However, more recent studies revealed significant intraspecific variability within *Az. spinosum* by identifying a new ribotype B, which has a fundamentally different toxin profile consisting of AZA-11 and -51 [26]. Moreover, another *Az. spinosum* ribotype (assigned as ribotype C) identified from the Argentinean shelf does not contain any AZA [27]. Toxigenic cells of ribotype B are of special concern, as sequence data and actual testing show that such strains are not quantitatively captured by the current *Az. spinosum* qPCR assay [26,28], whose design was based on ribotype A strains [29]. Considering the large number of amphidomatacean species known today and the high intraspecific variability of gene sequences and toxin production potential [27,30,31], it is remarkable that knowledge of the diversity in Irish waters is based on field sample records of *Az. caudatum* (Halldal) Nézan and Chomérat [32] and two strains of Amphidomataceae only, i.e., one strain of *Az. spinosum* [18] and one strain of *Am. languida* [33]. However, detailed knowledge on the local species inventory is important to identify other yet unknown sources of AZA and/or to evaluate the potential of local non-toxigenic species/strains for false positive signals either in light microscopy (LM) based and/or PCR based monitoring programs. Moreover, potential intraspecific variability of the presumably most important Irish AZA-producer, *Az. spinosum*, is completely unknown for Irish waters at present. Therefore, in summer 2018, a research survey in the Celtic Sea, in Irish coastal waters and in the North Sea was undertaken. The specific focus of this survey was to increase knowledge about the diversity and distribution of Amphidomataceae and their respective toxins in Irish coastal waters and in the North Sea. Field data of this survey, including qPCR-based abundance and distribution of toxigenic amphidomatacean species and their toxins, are presented elsewhere [34]. Next to field-sample data, on-board cell isolation and establishment of a large number of clonal amphidomatacean strains aimed at a better description of amphidomatacean diversity in the area. From 113 successfully isolated new strains from the survey, the non-toxigenic strains of four species (two of them new species) are presented elsewhere [35]. The focus of the present paper is to present detailed morphological, phylogenetic, and toxinological characterizations of 82 new strains representing the potential AZA producers *Az. spinosum* and *Am. languida*.

## 2. Materials and Methods

### 2.1. Field Work 

#### 2.1.1. Sampling

Samples were collected during the survey (HE-516) on-board RV Heincke between 17 July and 15 August 2018 covering the South- and West coast of Ireland and the North Sea (for a full list of stations see Wietkamp et al. [34]; CTD data are stored in Pangaea [36]). At each station, plankton samples were collected with 10 L Niskin bottles at 3 m, 10 m, and the depth-chlorophyll-maximum (DCM) layer. Five liters of seawater from each depth were filtered through a 20 μm mesh-size Nitex sieve, pooled, and well mixed.

#### 2.1.2. On Board Microscopy

Mixed bottle samples were used for on-board microscopical observation and documentation of live cells of Amphidomataceae. One-liter samples were pre-screened (20 μm Nitex mesh), gently concentrated by gravity filtration using a 3 μm polycarbonate filter (47 mm diameter, GE Healthcare, Little Chalfont, UK), and examined using an inverted microscope (Axiovert 200M, Zeiss, Göttingen, Germany). Cells of *Azadinium* and/or *Amphidoma* were pre-identified at high magnification (640×) based on general cell size and shape, on the presence of a theca, and on the presence of a distinctly pointed apex. Cells of interest were photographed with a digital camera (Axiocam MRc5, Zeiss).

#### 2.1.3. On-Board Isolation and Culture

Pre-identified cells of Amphidomataceae detected during the on-board live sample observations were isolated by micro-capillary into wells of 96-well plates filled with 0.2 mL filtered seawater from the sampling site. Plates were incubated at 15 °C under a photon flux density of approx. 50 μmol m^−2^ s^−1^ on a 16:8 h light:dark photocycle in a controlled environment growth chamber (Model MIR 252, Sanyo Biomedical, Wood Dale, IL, USA).

### 2.2. Characterization of Amphidomataceae Strains

#### 2.2.1. Culture Growth, Sampling, and Extraction

Isolation plates from the cruise were inspected after two weeks using a stereomicroscope (SZH-ILLD, Olympus, Hamburg, Germany) for the presence of *Azadinium*-like cells as inferred from the typical size, shape, and swimming behavior. From each positively identified well, a clonal strain was established by isolation of single cells via micro-capillary, and established cultures were thus clonal but not axenic. The clonal cultures were maintained in 70 mL plastic culture flasks at 15 °C in a natural seawater medium prepared with sterile-filtered (0.2 μm VacuCap filters, Pall Life Sciences, Dreieich, Germany) Antarctic seawater (salinity: 34, pH adjusted to 8.0) and enriched with 1/10 strength K-medium [37]; slightly modified by omitting the addition of ammonium ions.

For DNA extraction, each strain was grown in 70 mL plastic culture flasks at 15 °C under a photon flux density of 70 μmol m^−2^ s^−1^ on a 16:8 h light:dark photocycle. Fifty mL of healthy and growing culture (based on stereomicroscopic inspection of the live culture) were harvested by centrifugation (Eppendorf 5810R, Eppendorf, Hamburg, Germany; 3220× *g*, 10 min). Each pellet was transferred to a microtube, again centrifuged (Eppendorf 5415; 16,000× *g*, 5 min), and stored frozen in 500 μL SL1 lysis buffer (provided by the DNA extraction kit) at −80 °C until DNA extraction.

For toxin analysis, strains were grown under the standard culture conditions described above. For each harvest, cell density was determined by settling Lugol’s fixed samples and counting >400 cells under an inverted microscope in order to calculate toxin cell quota. Densely grown strains (ranging from ca. 1−7 × 10^4^ cells mL^−1^) were harvested by centrifugation (Eppendorf 5810R) at 3220× *g* for 10 min of 50 mL subsamples. The cell pellet was resuspended, transferred to a microtube, centrifuged again (Eppendorf 5415, 16,000× *g*, 5 min), and stored frozen (−20 °C) until use. For a number of selected strains, growth and harvest procedures were repeated several times to yield high biomass for increased sensitivity of the toxin detection method. The total number of cells harvested for these strains is listed in Appendix A.

A number of selected strains of *Azadinium spinosum* were sampled and analyzed for their AZA profile several times at various points in time, in a period of up to 18 months after isolation to evaluate the temporal stability of the toxin profile.

Cell pellets were extracted with 0.5 mL acetone and were vortexed every 10 min during one hour at room temperature. Homogenates were centrifuged (Eppendorf 5810 R) at 15 °C and 3220× *g* for 15 min. Filtrates were then adjusted with acetone to a final volume of 0.5 mL. The extracts were transferred to a 0.45 μm pore-size spin-filter (Millipore Ultrafree, Millipore, Burlington, MA, USA) and centrifuged (Eppendorf 5415 R) at 800× *g* for 30 s, with the resulting filtrate transferred into a liquid chromatography (LC) autosampler vial for liquid chromatography-mass spectroscopy (LC-MS/MS) analysis.

#### 2.2.2. Microscopy

Light Microscopy (LM) observation of living or fixed cells was carried out with an inverted microscope (Axiovert 200 M, Zeiss) or a compound microscope (Axioskop 2, Zeiss) by recording videos using a digital camera (Gryphax, Jenoptik, Jena, Germany) at full-HD resolution. Single frame micrographs were extracted using Corel Video Studio software (Version X8 pro). The shape and location of the nucleus was determined after staining of formalin-fixed cells with 4′-6-diamidino-2-phenylindole (DAPI, 0.1 μg mL^−1^ final concentration) for 10 min. Cell length and width were measured at 1000× microscopic magnification using Zeiss Axiovision software (Zeiss) and photographs of formaldehyde-fixed cells (1% final concentration) of strains growing at 15 °C taken with a digital camera (Axiocam MRc5, Zeiss).

For scanning electron microscopy (SEM), cells were collected by centrifugation (Eppendorf 5810R; 3220× *g*, 10 min) of 15 mL of culture. The supernatant was removed and the cell pellet re-suspended in 60% ethanol in a 2 mL microtube for 1 h at 4 °C to strip off the outer cell membrane. Subsequently, cells were pelleted by centrifugation (Eppendorf 5415 R, 16,000× *g*, 5 min) and resuspended in a 60:40 mixture of deionized water and seawater for 30 min at 4 °C. After centrifugation and removal of the diluted seawater supernatant, cells were fixed with formaldehyde (2% final concentration in a 60:40 mixture of deionized water and seawater) and stored at 4 °C for 3 h. Cells were then collected on polycarbonate filters (Millipore, 25 mm Ø, 3 μm pore-size) in a filter funnel where all subsequent washing and dehydration steps were carried out. A total of eight washings (2 mL deionized water each) were followed by a dehydration series in ethanol (30%, 50%, 70%, 80%, 95%, and 100%; 10 min each). Filters were dehydrated with hexamethyldisilazane (HMDS), first in 1:1 HMDS:EtOH followed by two times 100% HMDS, and then stored under gentle vacuum in a desiccator. Finally, filters were mounted on stubs, sputtercoated (SC500, Emscope, Ashford, UK) with gold-palladium and viewed under a scanning electron microscope (Quanta FEG 200, FEI, Eindhoven, The Netherlands). Some SEM micrographs were presented on a black background using Adobe Photoshop 6.0 (Adobe Systems, San Jose, CA, USA).

#### 2.2.3. Molecular Phylogeny

##### PCR Amplification and DNA Sequencing

The cell pellets for DNA extraction were collected in individual bead tubes together with 500 μL of the SL1 lysis buffer, both provided by the NucleoSpin Soil DNA extraction kit (Macherey and Nagel, Düren, Germany). The DNA extraction followed the manufacturer’s instructions, with slight variation. The bead tubes were not vortexed but shaken for 45 s and another 30 s at a speed of 4.0 m s^−1^ in a cell disrupter (FastPrep FP120, Thermo-Savant, Illkirch, France). For DNA elution, 2 × 50 μL of the provided elution buffer were used (to a final elution volume of 100 μL) to maximize the overall DNA yield. DNA was stored at −20 °C until further processing.

Sanger Sequencing of strain DNA was performed for the small subunit (SSU), the Internal Transcribed Spacer region (ITS1, 5.8S rRNA, ITS2) and the D1/D2 region of the large subunit (LSU), using the following primer sets: 1F (5′-AAC CTG GTT GAT CCT GCC AGT-3′) and 1528R (5′-TGA TCC TTC TGC AGG TTC ACC TAC-3′) for SSU [38]; ITSa (5′-CCA AGC TTC TAG ATC GTA ACA AGG (ACT)TC CGT AGG T-3′) and ITSb (5′-CCT GCA GTC GAC A(GT)A TGC TTA A(AG)T TCA GC(AG) GG-3′) for ITS [39]; DirF (5′-ACC CGC TGA ATT TAA GCA TA-3′) and D2CR (5′-CCT TGG TCC GTG TTT CAA GA-3′) for LSU [40]).

One part of the final sequences was gained by sending extracted DNA and primers to Eurofins sequencing facilities (Eurofins Genomics, Ebersberg, Germany), where sequences were generated on an ABI 3730 XL sequencer (Applied Biosystems by Thermo Fisher Scientific, Waltham, MA, USA) according to internal sequencing procedures.

The second part of the sequences was generated at the Alfred-Wegener-Institute (Bremerhaven, Germany). Each PCR reaction contained 16.3 μL ultra-pure H_2_O, 2.0 μL HotMaster Taq buffer (5Prime, Hamburg, Germany), 0.2 μL dNTPs (10 μM), 0.2 μL of each primer (10 μM), 0.1 μL of Taq Polymerase (Quantabio, Beverly, MA, USA) and 1.0 μL of extracted DNA template (10 ng μL^−1^) to a final reaction volume of 20 μL. PCR was conducted in a Nexus Gradient Mastercycler (Eppendorf) with the following condition: SSU amplification was performed according to the following settings: initialization at 94 °C for 5 min; 30 cycles of 94 °C for 2 min, 55 °C for 2 min, 68 °C for 3 min, and a final extension at 68 °C for 10 min. For ITS amplification, the settings were: 4 min at 94 °C, followed by 10 cycles of 50 s at 94 °C, 40 s at 58 °C, 1 min at 70 °C, and then 30 cycles of 45 s at 94 °C, 45 s at 50 °C, 1 min at 70 °C, and a final extension of 5 min at 70 °C.

LSU amplification: 2 min at 94 °C, followed by 30 cycles of 30 s at 94 °C, 30 s at 55 °C, 2 min at 65 °C, and a final extension of 10 min at 65 °C. The PCR amplicons were checked on a 1% agarose gel (in TE buffer, 70 mV, 30 min) to verify the expected length. The PCR amplicon was purified using the NucleoSpin Gel and PCR clean-up kit (Macherey-Nagel) and sequenced directly in both directions on an ABI PRISM 3730XL (Applied Biosystems by Thermo Fisher Scientific) as described in Tillmann et al. [41]. Raw sequence data were processed using the CLC Genomics Workbench 12 (Qiagen, Hilden, Germany).

##### Phylogenetic Analyses

Newly obtained SSU, ITS1-5.8S-ITS2 and/or partial LSU rRNA gene sequences were incorporated into available *Amphidoma, Azadinium,* and closely related sequences in GenBank. GenBank accession numbers are listed in Appendix A. Concatenated sequences were aligned using MAFFT v7.110 [42] online program (http://mafft.cbrc.jp/alignment/server/). Alignments were manually checked with BioEdit v. 7.0.5 [43]. Completed alignments of ITS1-5.8S-ITS2 sequences were imported into PAUP *4b10 software [44] to estimate divergence rates using simple uncorrected pairwise (p) distance matrices. The secondary structures of ITS2 sequences of five strains of *Az. spinosum* or *Az.* cf. *spinosum* were predicted using the Mfold program [45] (http://mfold.rit.albany.edu/?q=mfold/RNA-Folding-Form).

For Bayesian inference (BI), the program jModelTest [46] was used to select the most appropriate model of molecular evolution with Akaike Information Criterion (AIC). Bayesian reconstruction of the data matrix was performed using MrBayes 3.2 [47] with the best-fitting substitution model (GTR+G). Four Markov chain Monte Carlo (MCMC) chains ran for 10,000,000 generations, sampling every 1000 generations. The convergence of the MCMC chains was examined in TRACER 1.7 [48], and the first 10% of the samples were discarded as ‘burn-in’, well after stationarity had been reached. A majority rule consensus tree was created in order to examine the posterior probabilities of each clade. Maximum likelihood (ML) analyses were conducted with RaxML v7.2.6 [49] on the T-REX web server [50]. Data were analyzed using the GTR+G approximation and the rapid hill-climbing algorithm was used. Node support was assessed with 1000 bootstrap replicates.

##### qPCR Assay Specificity of Newly Obtained Strains

Newly obtained strain DNA sequences were aligned with the primer and probe sequences of the current *Az. spinosum* qPCR assay using MEGA7 [51] to look in silico for base pair (bp) mismatches, which potentially affect the assay specificity.

Subsequently, DNA of newly obtained *Az. spinosum* ribotype A (4-F8, 5-C11, 6-G8), ribotype B (5-F3, 8-B8), *Az.* cf. *spinosum* (1-H10, 2-A3, 5-B9, 5-D3, 6-A1) and *Am. languida* (5-F11, 8-D10) strains was subjected to in vitro specificity testing with the current qPCR assays for *Az. spinosum*, *Az. poporum* and *Am. languida* following the detailed descriptions in Wietkamp et al. [34].

#### 2.2.4. Chemical Analysis of Azaspiracids

Extracts of strains were screened for known AZA in the selected reaction monitoring (SRM) mode with an analytical system consisting of triple quadrupole mass spectrometer (API 4000 QTrap, Sciex, Darmstadt, Germany) equipped with a TurboSpray interface coupled to LC equipment (model LC 1100, Agilent, Waldbronn, Germany) that included a solvent reservoir, inline degasser (G1379A), binary pump (G1311A), refrigerated autosampler (G1329A/G1330B), and temperature-controlled column oven (G1316A). Separation of AZA (5 μL sample injection volume) was performed by reverse-phase chromatography on a C8 phase. The analytical column (50 × 2 mm) was packed with 3 μm Hypersil BDS 120 Å (Phenomenex, Aschaffenburg, Germany) and maintained at 20 °C. The flow rate was 0.2 mL min^−1^, and gradient elution was performed with two eluents, where eluent A was water, and eluent B was acetonitrile/water (95:5 *v*/*v*), both containing 2.0 mM ammonium formate and 50 mM formic acid. Initial conditions were 8 min column equilibration with 30% B, followed by a linear gradient to 100% B in 8 min and isocratic elution until 18 min with 100% B then returning to initial conditions until 21 min (total run time: 29 min). AZA profiles were determined in the SRM mode in one period (0–18) min with curtain gas: 10 psi, CAD: medium, ion spray voltage: 5500 V, temperature: ambient, nebulizer gas: 10 psi, auxiliary gas: off, interface heater: on, declustering potential: 100 V, entrance potential: 10 V, exit potential: 30 V. SRM experiments were carried out in positive ion mode by selecting the transitions shown in Appendix A
Appendix A.

In addition, precursor ion experiments were performed. Precursors of the characteristic AZA fragments *m/z* 348, *m/z* 350, *m/z* 360, *m/z* 362, and *m/z* 378 were scanned in the positive-ion mode from *m/z* 500 to 1000 under the following conditions: curtain gas, 10 psi; CAD, medium; ion spray voltage, 5500 V; temperature, ambient; nebulizer gas, 10 psi; auxiliary gas, off; interface heater, on; declustering potential, 100 V; entrance potential, 10 V; collision energy, 70 V; exit potential, 12 V. Collision induced dissociation (CID) spectra of the *m/z* values 716, 830, 842, 856, 858, and 872 were recorded in the Enhanced Product Ion (EPI) mode in the mass range from *m/z* 150 to 930. Positive ionization and unit resolution mode were used. The following parameters were applied: curtain gas: 10 psi, CAD: medium, ion spray voltage: 5500 V, temperature: ambient, nebulizer gas: 10 psi, auxiliary gas: off, interface heater: on, declustering potential: 100 V, collision energy spread: 0, 10 V, and collision energy: 70 V, exit potential, 12 V.

### 2.3. Statistics

Data of AZA cell quota and ratios were plotted using the box-whisker plot option of Microsoft Excel using the median, and first and third quartile, and plotting all data points. Outliers were defined as outside 1.5× interquartile range. AZA cell quota and AZA ratios were tested for normal distribution by Shapiro–Wilk tests. Based on these results, all AZA cell quota and ratios were tested by One-Way-ANOVA on ranks (Kruskal–Wallis test). Statistical testing was performed using Statistica (version 9.1, StatSoft, Tulsa, OK, USA).

## 3. Results

A total of 113 new strains of Amphidomataceae was obtained from the Irish coast and from the central North Sea (Table 1). Of those, a total of 31 non-toxigenic strains representing *Az. caudatum* var. *margalefii* (Rampi) Nézan and Chomérat (one strain), a taxon identified as *Az.* cf. *zhuanum* Z. Luo, H. Gu and Tillmann (one strain) as well as two new species, *Az. galwayense* Salas and Tillmann (three strains) and *Az. perfusorium* Tillmann and Salas (26 strains), are reported elsewhere [35]. The remaining 82 new strains representing potentially toxigenic *Az. spinosum* and *Am. languida* strains (Table 1), are reported here. Within strains designated as *Az. spinosum* we identified strains of toxigenic ribotypes A and B, as well as five new strains without detectable AZA. The latter revealed significant nucleotide differences compared to known *Az. spinosum* sequences and therefore also clustered apart from other *Az. spinosum* strains within the phylogenetic tree. These strains are subsequently designated as *Az.* cf. *spinosum*.

Strains were obtained from the Irish coast and from the central North Sea (Figure 1), with multiple strains of *Az. spinosum* obtained from stations 35, 45, and 71. All five *Az.* cf. *spinosum* strains originated from station 35, and new strains of *Am. languida* exclusively originated from the central North Sea station 71 (Figure 1). All strains were analyzed for their AZA profiles, most were investigated morphologically using LM or SEM, and sequence data were obtained for a selected number of strains (a detailed compilation of information on each strain can be found in Appendix A).

### 3.1. Phylogeny of Strains

All 24 *Az. spinosum* strains producing AZA-1 (as well as AZA-2 and AZA-33) for which sequence data were generated shared identical LSU and ITS rDNA sequences except for two strains, which showed one bp difference in LSU sequences. Likewise, all seven *Az. spinosum* strains producing AZA-11 (and AZA-51) shared identical LSU and ITS rDNA sequences. The two different *Az. spinosum* groups differed from each other in LSU by 7 bp and in ITS by 11 bp. The five non-toxigenic *Az.* cf. *spinosum* strains differed from each other in LSU by 6 bp. In ITS, four of them shared identical sequences but differed from strain 5-B9 by 13 bp. In contrast, all strains of *Am. languida* shared identical sequences.

Genetic distances were less than 0.04 among *Az. spinosum* ribotypes A, B, and C, but varied from 0.05 to 0.07 when compared with *Az.* cf. *spinosum* sequences. Moreover, relatively low ITS genetic distances were calculated between *Az.* cf. *spinosum* and *Az. obesum* Tillmann and Elbrächter (0.03–0.04). Lower p-distances were also seen between *Az. obseum* and *Az. trinitatum* Tillmann and Nézan (0.05–0.06), or *Az. poporum* (0.05) (Appendix A).

The ITS2 secondary structure of five strains, representing ribotypes A, B, C, and *Az.* cf. *spinosum* (two strains), was predicted. All of them showed four main helices (I, II, III, IV) but the number of loops in helices II and III varied markedly. There was one compensatory base change (CBC, compensatory base change on both sides of a helix pairing) in helix IV between ribotypes A/B/C and *Az.* cf. *spinosum* strains (Table 2, Appendix A).

The maximum likelihood (ML) and Bayesian inference (BI) analysis based on concatenated SSU, ITS-5.8S, and partial LSU rRNA gene sequences yielded similar phylogenetic trees. The BI tree is illustrated in Figure 2. The family Amphidomataceae was well resolved with maximal support (1.0 Bayesian posterior probability (BPP) and 100 bootstrap support (BS)) consisting of two clades. The first clade consisted of *Amphidoma parvula* Tillmann and Gottschling and *Am. languida* with maximal support. New strains of *Am. languida* grouped together with strains from elsewhere with maximal support. The second clade comprised all *Azadinium* species but received low support. Two *Azadinium* species (*Az. concinnum* Tillmann and Nézan and *Az. perforatum* Tillmann, Wietkam and H.Gu) diverged early and formed a sister clade with the remaining *Azadinium* species, which formed a monophyletic group with maximal support. *Azadinium spinosum* consisted of three ribotypes with maximal support. Ribotype A included strains 6-A10, 4-G9, 3-B4, and 5-C11 with low BPP but high BS (98). Ribotype B included strains 5-G8 and 7-D3 with strong support (0.96 BPP/100 BS). The five non-toxigenic *Az.* cf. *spinosum* strains were well resolved and consisted of two clades. One of them comprised strains 6-A1, 5-D3, 2-A3, 1-H10, and another comprised strain 5-B9. They were closest to *Az. obesum* with low support.

#### qPCR Assay Specificity with Newly Obtained Strains

In silico specificity checking for newly obtained *Az. spinosum* ribotype A strains did not show any base pair mismatches with the primers and probe of the current *Az. spinosum* qPCR assay. Sequences of the new ribotype B strains revealed one bp mismatch with the probe and two bp mismatches with the reverse primer. *Azadinium* cf. *spinosum* sequences had three bp mismatches with both the probe and the reverse primer (Table 3).

In vitro testing of new *Az. spinosum* ribotype A strains showed the same amplification efficiency (C_T_ = 18.4 for strains 4-F8 and 5-C11, C_T_ = 19.4 for strain 6-G8) as the ribotype A reference strains (Appendix A). DNA of the newly obtained ribotype B strains was amplified with less efficiency compared to the ribotype A DNA. However, the same efficiency (C_T_ = 25.6 for strain 5-F3 and C_T_ = 25.4 for strain 8-B8) was observed compared to the ribotype B reference strains. None of the newly isolated *Az.* cf. *spinosum* strains was amplified in the *Az. spinosum* qPCR assay. DNA of the newly obtained *Am. languida* strains showed the same amplification efficiency (C_T_ = 20.7 for strains 5-F11, C_T_ = 20.4 for strain 8-D10) as the reference strains (Appendix A).

Application of all tested *Az. spinosum*, *Az.* cf. *spinosum* and *Am. languida* strains did not reveal any detectable false-positive amplifications (Appendix A). Limit of detection (LOD) was 0.1 pg target DNA μL^−1^ for all three qPCR assays.

### 3.2. Morphological Identification

Morphological identification of the strains conformed their phylogenetic placement and morphology of *Az. spinosum* (ribotypes A and B), *Az.* cf. *spinosum* and *Am. languida* are shown below.

#### *Azadinium spinosum* 

Ribotype A: All strains assigned to ribotype A in our phylogenetic placement were similar in size (Appendix A), shape, and general appearance (Figure 3A–D,G,H). Cells of all *Az. spinosum* ribotype A strains consistently had an antapical spine (Figure 3A,B,G,H,L). One large pyrenoid with a starch sheath (visible as a ring-like structure) was located in the episome (Figure 3B–D). The nucleus was generally round to slightly ellipsoid but could be, presumably in the early stages of cell division, more elongated as well (Figure 3E,F). Cells of all strains which were examined with SEM (Appendix A) had a conspicuous ventral pore on the left suture of plate 1′ (Figure 3G,I) and the thecal plate pattern of epi- and hypotheca typical for the species (Figure 3K,L). Moreover, cells consistently had a distinct rim around the pore plate (Figure 3J,K).

Ribotype B: All ribotype B strains were also similar in size, shape, and general appearance (Figure 4 for strain 5-F6, Figure plates for other ribotype B strains are Appendix A). With respect to most morphological features such as the presence and location of a pyrenoid (Figure 4A), presence of the antapical spine (Figure 4B,G–I), the thecal plate pattern of epi- and hypotheca (Figure 5), and location of the ventral pore (Figure 4G,H; Figure 5A,C), ribotype B strains were not distinguishable from ribotype A strains. The nucleus was posterior in position and ellipsoid and elongated in most cases (Figure 4D–F). In contrast to ribotype A strains, for all seven strains identified as ribotype B, a distinct rim around the pore plate was missing (Figure 5A,E).

*Azadinium* cf. *spinosum*: Morphology of the *Az.* cf. *spinosum* strains is compiled in Figure 6 and Figure 7 for strain 5-B9 (Figure plates for other *Az.* cf. *spinosum* strains can be found in Appendix A). In terms of morphology, these strains shared the same morphological features described as distinctive for *Az. spinosum*, i.e., possession of one prominent pyrenoid in the episome (Figure 6A,F,G), an antapical spine (Figure 6E,F,K–M), a roundish posterior nucleus (Figure 6H,I) that can be elongated during cell division (Figure 6J), and a ventral pore (vp) located on the left suture of plate 1′ (Figure 6K,L and Figure 7D–G). Plate pattern of epi- and hypotheca (Figure 7A,B) as well as of the cingulum and sulcus (Figure 7H–J) were indistinguishable from other *Az. spinosum* strains. Cells of all *Az.* cf. *spinosum* strains had a distinct rim around the pore plate (Figure 7A,C–G).

*Amphidoma languida*: All ten new *Am. languida* strains from the survey were obtained from the central North Sea station 71. They all shared an identical morphology as observed in LM (Figure 8). In accordance with the species description, cells consistently had one large pyrenoid with a starch sheath (visible as a ring-like structure) located in the episome (Figure 8B,C). Detailed SEM (Figure 8D–M) performed for a selected number of strains (Appendix A) revealed the Kofoidian plate pattern for the species (pore plate (Po), cover plate (cp), X plate or canal plate (X), 6′, 0a, 6″, 6C, 5S, 6″′, 2⁗) (Figure 8F,G,K,L), a vp located at the right side of plate 1′ close to the pore plate (Figure 8F,I,J), and a large antapical pore located on the second antapical plate (Figure 8E,G,H). A number of, but not all, cells in the clonal cultures had a round ventral depression located at the anterior tip of the anterior sulcal plate (Figure 8M).

### 3.3. Toxins

Azaspiracid profiles obtained for all strains initially referred to as *Az. spinosum* revealed the presence of three groups of strains. The first group corresponding to ribotype A was characterized by the presence of AZA-1, AZA-2, and AZA-33, whereas the second group corresponding to all ribotype B strains was determined by the presence of AZA-11 and AZA-51. The third group consisted of all *Az.* cf. *spinosum* strains and lacked any AZA. The limit of detection for known AZA congeners and for yet unknown AZA (based on precursor experiments) were estimated for a number of selected strains of each group using high-biomass samples and are reported in Appendix A.

AZA profiles differed between strains of ribotype A. There were four different combinations of AZA-1, AZA-2, and AZA-33: (1) the majority (32 strains) had all three AZA congeners, (2) 10 strains contained only AZA-1, (3) 13 strains contained AZA-1 and AZA-2 but lacked AZA-33, and (4) five strains contained only AZA-1 and AZA-33 and lacked AZA-2. In contrast, no AZA profile variability was observed in group 2 (ribotype B) where all seven strains contained both AZA-11 and AZA-51.

As shown for a number of selected strains, all distinct toxin profiles (including absence of AZA of *Az.* cf. *spinosum* strains) were consistent and stable over time, as estimated for a period of up to 18 months (Table 4).

However, in quantitative terms, AZA cell quotas were highly variable. Including all strains and all repeated analyses for single ribotype A strains, total AZA cell quota (sum of all detected AZA) varied 53-fold and ranged from 1.2 to 63.1 fg cell^−1^ (Figure 9). Each single AZA compound showed high variability as well, with AZA-2 showing the highest fold-change of 330. Ratios of AZA compounds were also quite variable. The median ratio of AZA-1 to AZA-2 or AZA-33 was 2.2 and 5.0, respectively, but for single strains/single analysis ratios <1 were also obtained, and the same was observed for AZA-2/AZA-33 ratios (Figure 9). Cell quotas of all single AZA congeners and AZA-ratios were tested for significant differences between the four different toxin profile groups (Table 5, Appendix A). Kruskal–Wallis tests revealed significant differences for total AZA cell quotas (H = 9.81, *p* = 0.020), whereas AZA-1, AZA-2, and AZA-33 were not significantly different between the four toxin-profile groups of ribotype A strains (*p* > 0.3). Whereas the AZA-1/AZA-2 ratio was just slightly below the 0.05 significance level (H = 3.68, *p* = 0.055) the AZA-1/AZA-33 ratios were not significantly different (*p* = 0.865) (Table 5).

High variability in AZA cell quota was also obvious for those ribotype A strains for which multiple independent time-series analyses were available (Figure 10, summary statistic tables are listed in the Appendix A). Total AZA within a single ribotype A strain varied up to 16-fold (strain 5-E4), but for other strains (e.g., 3-E6, 4-E11) cell quota was quite consistent. Fold-changes of multiple analyses of AZA-1/AZA-2 ratios were <2 for many strains (Figure 10, Appendix A), but other strains showed high (up to 6.5-fold) changes of this ratio. Kruskal–Wallis ANOVA revealed that total AZA cell quota were not different between ribotype A strains (H = 26.37, *p* = 0.193). In contrast, for all AZA ratios (1/2, 1/33, 2/33), there were highly significant differences between strains (*p* < 0.01) (Table 5).

Toxin profile of *Az. spinosum* ribotype B strains consisted of AZA-11 and AZA-51 and was constant over time (Table 4). Total AZA cell quotas for ribotype-B strains ranged from <0.1 to 14.0 fg cell^−1^ (Figure 11). While median cell quotas of all analyses were similar for AZA-11 and AZA-51, the AZA-11/AZA-51 ratio (with a median value of 1.0) of individual analysis ranged from 0.1 to 2.3 (Figure 11). Total AZA cell quota estimates of single strains were variable as well, with fold changes of multiple estimates ranging up to 68-fold (Figure 12, summary statistics are listed in the Appendix A). The ratio of AZA-11/AZA-51 within single strains also varied around 1.0 but was consistently <1.0 or >1.0 for two or one strain, respectively (Figure 12). For none of the AZA parameters (total AZA cell quota, AZA-11, AZA-51, ratio 11/51) there were statistical differences between strains (Kruskal–Wallis ANOVA, *p* > 0.07) (Table 5).

All strains of *Am. languida* produced AZA-38 and AZA-39 (for LOD of other known and/or unknown AZA compounds, see Appendix A). Total AZA cell quota ranged about 100-fold from 0.3 to 29.6 fg cell^−1^ (Figure 13). The median ratio of AZA-38/-39 was slightly below 1, but the ratio of single analyses varied between 0.5 and 1.4 (Figure 13). There were no repeated analyses of single strains, so temporal stability and variability in cell quota for *Am. languida* could not be assessed.

## 4. Discussion

Isolation and characterization of multiple amphidomatacean strains from Irish waters and the North Sea revealed a number of new insights into the diversity of this group of potentially toxic microalgae. At the taxonomic level, we identified and described two new and non-toxigenic *Azadinium* species presented in detail elsewhere [35]. In addition, we add a number of important facts about the diversity of the known toxigenic species *Az. spinosum* and *Am. languida* and thereby have added complementary information to the field data set on the abundance and distribution of toxigenic Amphidomataceae and their toxins to that which have been published previously [34].

*Azadinium spinosum*, the first identified source organism of AZA, is regarded as the most important AZA producer in Irish waters [18,25,34]. The first strains of this species isolated from Scotland [21], Denmark [52], the Shetland Islands [24], and Ireland [18] all share the same sequence data and toxin profile consisting of AZA-1, AZA-2, and AZA-33, but subsequent multiple strain studies from Norway and Argentina revealed ribotype divergence and toxin profile diversity within *Az. spinosum* [26,27]. Ribotype B strains from Norway mainly have AZA-11 and AZA-51, whereas a single ribotype B strain from Argentina only produce AZA-2. All ribotype B strains also differ morphologically from ribotype A strains by lacking a distinct bulged rim around the apical pore plate [26,27]. Moreover, there is morphological variability within ribotype B: The single AZA-2 producing strain from Argentina has a striking slender shape and the first epithecal intercalary plate 1a has contact with the first apical plate, a character stage that is different from all other strains of *Az. spinosum*, irrespective of the ribotype [27]. All ribotype C strains are morphologically indistinguishable from the type strain exhibiting ribotype A, but consistently lack any AZA. So far, in *Az. spinosum* there is thus found: (1) AZA strain profile variability among different ribotypes; (2) morphological differences within a distinct ribotype, and (3) minor but consistent morphological differentiation between different ribotypes. With the data presented here, we can confirm these findings and also show that there is toxin profile variability within *Az. spinosum* ribotype A.

### 4.1. Azadinium spinosum Ribotype A and B

These results presented here thus confirm the presence of two different *Az. spinosum* ribotypes (A and B) in the North Atlantic and their different AZA profiles (dominated by AZA-1, AZA-2, and AZA-11, AZA-51, respectively). From a chemical point of view, it has to be pointed out that AZA-11 and AZA-51 of ribotype B are 3-hydroxylated, whereas AZA of ribotype A does not have any substituents at C3. Moreover, the major AZA of both ribotypes, i.e., AZA-1/AZA-2 of ribotype A and AZA-11/-AZA-51 of ribotype B, consist of a methylated and unmethylated pair: AZA-2 is a methylated form of AZA-1 and AZA-11 is a methylated form of AZA-51. However, despite this common feature, there is a difference between both ribotypes, which is the methylation site. AZA-1 of ribotype A is missing methylation at C8, whereas AZA-51 of ribotype B is missing methylation at C24 (Figure 14). Interestingly, a similar pair of AZA congeners (3-hydroxylated or not) between different ribotypes can be seen within *Az. poporum*, where strains from Argentina (ribotype C2, see [53]) have AZA-2 (lack of hydroxylation at C3) whereas *Az. poporum* ribotype A1 (see [53]) strains from Chile solely having AZA-11 and -62 (3-hydroxylated) [41,54,55].

*Azadinium spinosum* has been reported to contain several additional compounds next to the above mentioned major AZAs, such as AZA-33 [26], AZA-2 methyl ester [54], phosphorylated forms of AZA-1 and AZA-2 [26,54], and AZA-34 and AZA-35 [56]. However, a number of minor AZA in *Az. spinosum* culture have been identified in dense stationary phase cultures only and thus are likely to be products of bacterial/chemical AZA degradation and not directly produced by the dinoflagellates [57]. Minor azaspiracids identified in the present study in ribotype A strains also include methylated AZA-1 and AZA-2 as well as phosphorylated forms of both AZA, and in some analyses of ribotype B strains phosphorylated AZA-11 and AZA-51 were detected. All these minor compounds always occurred in low quantities (<1 fg cell^−1^) and were not always detected, but this is likely because the limit of detection prevented detection of these minor compounds in low-biomass samples.

The multi-strain comparison of ribotype A and B strains show that in most cases, ribotype A strains contained a higher cell quota of AZA than ribotype B (median of all ribotype B strains: 1.0 fg cell^−1^; compared to 9.3 fg cell^−1^ for ribotype A strains), indicating that ribotype A is more toxic than ribotype B. The same was found for A and B strains from the Norwegian coasts (median total AZA cell quota for ribotype B strains: 1.1 fg cell^−1^; median total AZA cell quota for ribotype A strains: 4.5 fg cell^−1^) [26]. However, it has to be kept in mind that within ribotype A strains, a low AZA cell quota can also be found (Figure 9), so it might be better to say that up to now, no high cell quota (>15 fg cell^−1^) has been measured within B strains but are regularly registered within ribotype A strains. However, specific toxicity is not yet known for AZA-11 and AZA-51, and thus, a direct conversion of cell quota to toxicity for a better comparison with AZA-1 and AZA-2 is not possible at the moment. One important finding related to *Az. spinosum* ribotypes A and B is that even with a large number of *Az. spinosum* strains isolated from Ireland (57 strains), no single ribotype B strain was obtained, whereas B strains were the majority (seven of ten) of strains isolated from the North Sea (Figure 15). This relative dominance of B strains in the North Sea was not accompanied by the detection of AZA-11 or AZA-51 in the North Sea [34], showing that *Az. spinosum* ribotype B strain density in the North Sea is not at a critical level. The dominance of B-type *Az. spinosum* in the North Sea, together with the lower qPCR quantification efficiency of B compared to A strains (Appendix A), may have contributed to the comparable lower total amphidomatacean density estimate based on qPCR in the North Sea compared to microscopy counts [34]. In any case, the snapshot in time obtained with the present survey does not allow to rule out the presence of B-type strains around Ireland, but our data at least indicate that current monitoring using the A-strain specific qPCR assay does not appear to be heavily biased.

### 4.2. Toxin Profile Variability of Ribotype A

A new finding of the present study is that there is toxin profile variability within ribotype A. All ribotype A strains produce AZA-1, and toxin profile variability within ribotype A strain only refers to the presence/absence of AZA-2 and/or AZA-33. However, the implication of this pattern in relation to AZA synthesis pathways is not clear. Strains with AZA-1 but without AZA-33 indicate that the reduced molecule size of AZA-33 (molecular mass of 715 Da) cannot be considered a precursor for AZA-1. In contrast to the ribotype A AZA profiles reported here, a single *Az. spinosum* ribotype B strain from Argentina lacks AZA-1 and only produces AZA-2 [27], and the production of solely AZA-2 is also known from *Azadinium poporum* [54].

Structural variability of AZA profiles within a ribotype thus seems to be a common feature of toxic Amphidomataceae as it has also been seen in *Az. poporum* [31,58,59] and *Az. spinosum* ribotype B strains [27]. In any case, the multiple strain approach clearly showed that within ribotype A strains there were consistent and stable (at least for one year) differences in presence/absence of AZA-1, AZA-2 and/or AZA-33. The implications of this, e.g., in terms of AZA synthesis pathways, are obscure at the moment but indicate that some AZA structural details may not be of vital importance for any physiological and/or ecological function. Ribotype A strains with deviating AZA profile all originate from Irish waters, and all three new *Az. spinosum* ribotype A strains from the North Sea had AZA-1, AZA-2- and AZA-33. However, among Irish strains, the presence of AZA-1, AZA-2, and AZA-33 was clearly the quantitatively dominant pattern (>50%; 32 of 60 strains), which may explain why this variability was not seen in previous Atlantic strains and not in the new North Sea strains either.

Notably, none of thirteen toxigenic strains tested after more than one year in culture completely lost AZA production potential. Moreover, our time series data provide no statistical support that toxin production may consistently diminish with time under cultivation. However, significant quantitative changes will be very difficult to be evaluated experimentally, considering that the underlying causes for the extraordinary large variability of AZA cell quota seen in the present study (Figure 9, Figure 11, and Figure 13) are poorly understood.

### 4.3. Cell Quota Variability

A striking result of the multiple strain comparison is the enormous variability of AZA cell quota. Previous experimental studies show that the cell quota of a given strain may change in response to environmental conditions like temperature [57,60,61] or nutrient conditions [62]. However, here we used the same growth conditions for all strains. Another factor affecting AZA cell quota is the growth stage; cells in stationary growth usually have higher cell quotas compared to the exponential phase [57,61,62], which might be explained by AZA accumulation when cell division stops. The growth phase of cultures used for toxin sampling in the present study was not fully controlled, so this may have contributed to the observed high fold differences among and within strains. Moreover, high variability (up to 20-fold differences in AZA-2 cell quota) is obvious in data presented by Li et al. [62] for different sets of experiments (under the same conditions) with *Az. poporum*, and ~ five-fold differences in cell quota of the same strain of *Az. spinosum* in different experiments performed at different times of the year are reported [57]. Such a huge variability in toxin cell quota, even growing the same strain under identical environmental conditions, clearly indicates that there are other factors that are difficult, if not impossible, to control. In this respect, almost nothing is known about potential rhythmic or seasonal cycles in toxin production or long-term changes in response to the artificial laboratory environment without competitive or food web interactions.

In conclusion, AZA cell quota estimates may vary considerably within a species (i.e., among strains) but also within a given strain and thus are of limited significance when simply extrapolating abundance data to evaluate toxic potential. However, the results presented here show that the qualitative toxin profile of a given strain is stable at least for one year in culture and thus likely to be genetically fixed. Nevertheless, long term loss of AZA production potential may occur, as unpublished information indicates [57].

### 4.4. Non-Toxigenic Az. cf. spinosum

Variability within *Az. spinosum* becomes even more complex considering the newly identified group of strains listed here as *Az.* cf. *spinosum*. This taxon conforms morphologically with *Az. spinosum*, but lacks AZA and groups in a phylogenetic cluster outside other *Az. spinosum*. A non-toxigenic *Az. spinosum* morphotype is present in Argentina, but these strains have been shown to cluster with other *Az. spinosum* forming a ribotype C clade. Due to the lack of morphological differences between *Az.* cf. *spinosum* and other *Az. spinosum* ribotypes A and C, and despite their significant sequence differences and presence of CBCs between *Az.* cf. *spinosum* and all *Az. spinosum* ribotypes (Table 2), we currently refrain to finally conclude on the taxonomic level of these new strains, which we designate here as *Az*. cf. *spinosum*. All five strains of *Az*. cf. *spinosum* originate from the same station, but this does not necessarily indicate a limited distribution. In any case, this non-toxigenic taxon is of importance for monitoring because, with LM and even SEM, it is impossible to differentiate between toxin-producing *Az. spinosum* and these non-toxigenic cells. It is thus important to point out that *Az*. cf. *spinosum* without AZA production differ significantly in the qPCR assay relevant sequence area and thus do not produce (false) positive signals in the *Az. spinosum* assay.

### 4.5. Amphidoma languida

In contrast to *Az. spinosum*, there was no toxin profile variability among the ten new *Am. languida* strains from the North Sea. However, it has to be kept in mind that all these strains were obtained from a fairly dense bloom population [34] from the same station. Toxin profile variability within *Am. languida* is known from a Spanish Atlantic strain, which contains AZA-2 and -43 [17]. No new *Am. languida* strains were obtained from Irish waters, but the specific qPCR assay indicates that the species is widely present in the area [34,63,64]. However, densities seem to be lower compared to *Az. spinosum*, which might explain the lack of new *Am. languida* strains from Irish coastal waters in this study. In terms of AZA cell quota, it is important to note that within *Am. languida*, there was the same high intraspecific variability (Figure 13) as reported in previous studies [26,53] and as seen for AZA cell quotas in *Az. spinosum*.

## 5. Conclusions

The approach of multi-strain isolation and characterization has enabled deeper insights into the molecular, morphological, and toxinological variability within Amphidomataceae, and has significantly added to our pre-existing knowledge of these species in Irish coastal waters and in the North Sea.

New details that have emerged from this study include: The magnitude of AZA cell quota variability observed, the previously unknown differences in AZA profile among *Az. spinosum* ribotype A strains, the presence and distribution of ribotype A and B in the area, and the identification of the non-toxigenic *Az.* cf. *spinosum*. These details may altogether help to better understand the AZA toxin profile in these areas and to explain the differences that are often observed in the Irish biotoxin and phytoplankton monitoring programs when trying to correlate AZA concentration in shellfish (via LC-MS/MS) with *Azadinium* cell abundances obtained with LM and/or qPCR.

Likewise, it is important to be aware of the current *Az. spinosum* qPCR assay is not addressing the diversity of strains/ribotypes in Irish and North Sea waters. New specific assay specifically targeting toxigenic ribotypes A or B, and also for the non-AZA ribotype C and/or the *Az.* cf. *spinosum* would help to more closely look at diversity and geographic distribution, and would also allow more focused alerts to the shellfish industry. This newly gathered information would also feed into predictive modeling and forecasting tools used by shellfish industries, monitoring agencies, and regulatory authorities when ascertaining the risk or likelihood of AZA accumulation in shellfish, and also to provide further clarity on predicting the trends and patterns observed in the onset and during AZA events which can lead to prolonged closures of shellfish production areas.

## Figures and Tables

**Figure 1 microorganisms-09-00134-f001:**
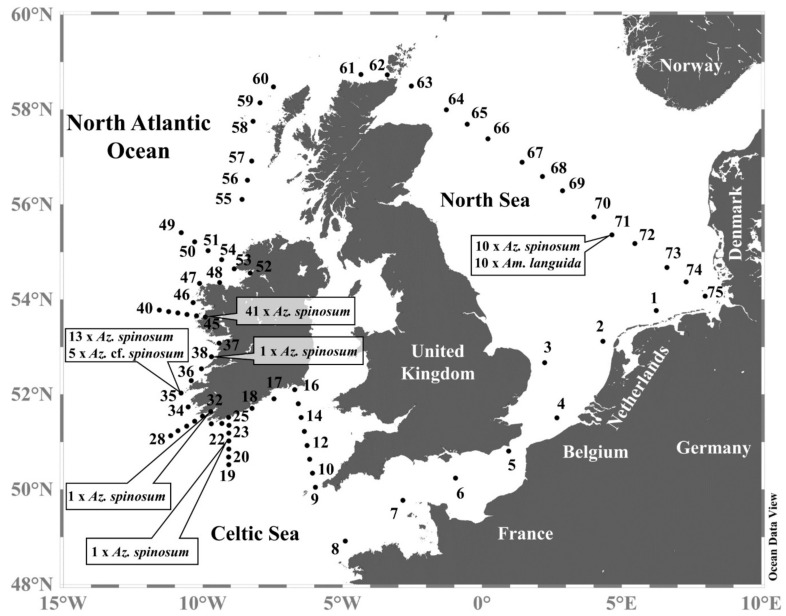
Map of the study area highlighting sample stations where *Azadinium* and *Amphidoma* strains were isolated.

**Figure 2 microorganisms-09-00134-f002:**
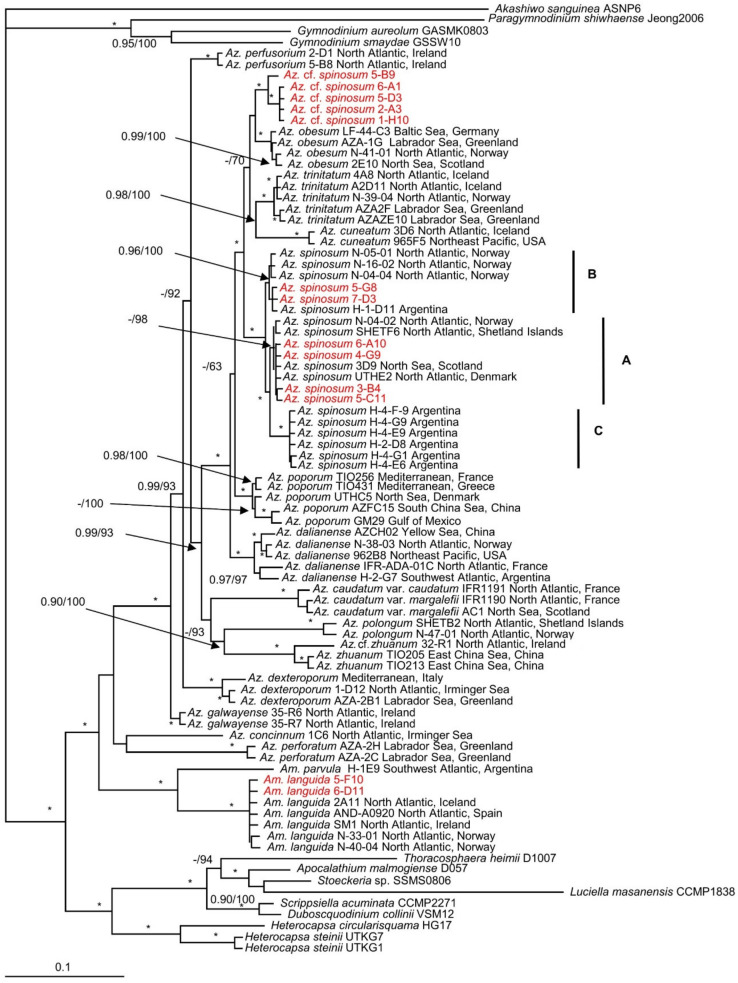
Molecular phylogeny of *Azadinium* and *Amphidoma* inferred from concatenated small subunit (SSU), internal transcribed region (ITS)-5.8S, and partial large subunit (LSU) rRNA gene sequences using Bayesian inference (BI). New sequences of *Azadinium spinosum*, *Az.* cf. *spinosum*, and *Amphidoma languida* are indicated in red. Ribotypes of *Az. spinosum* are marked with A, B, and C. Scale bar indicates the number of nucleotide substitutions per site. Numbers on branches are statistical support values (left, Bayesian posterior probabilities; right, maximum likelihood (ML) bootstrap support values). Bootstrap values >50% and posterior probabilities (pp) above 0.9 are shown. Asterisks (*) indicate maximal support (pp = 1.00 in BI and bootstrap support = 100% in ML, respectively).

**Figure 3 microorganisms-09-00134-f003:**
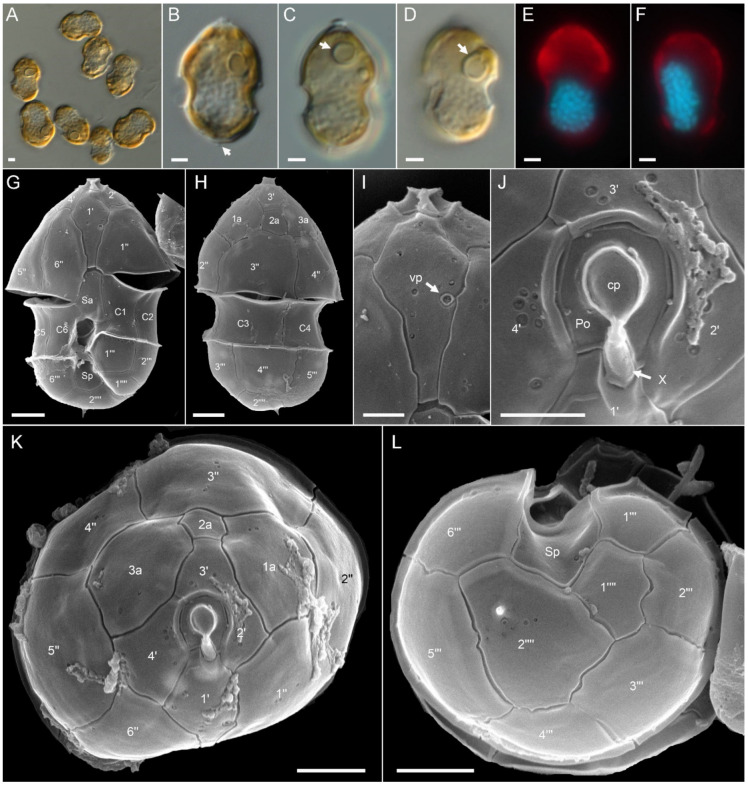
*Azadinium spinosum* ribotype A strains. (**A**–**D**) Light microscopy (LM) images of formalin fixed (**A**,**B**) or living (**C**,**D**) cells to indicate general size and shape. Note the antapical spine (arrow in **B**) and the distinct pyrenoid in the episome (arrows in **C**,**D**). (**E**,**F**) Formalin fixed and DAPI-stained cells viewed with UV excitation to indicate shape and position of the nucleus. (**G**–**L**) Scanning electron microscopy (SEM) images of different thecae. (**G**) Ventral view. (**H**) Dorsal view. (**I**) First apical plate in ventral view. Note the position of the ventral pore (vp). (**J**) Detailed view of the apical pore complex (APC). (**K**) Apical view of epithecal plates. (**L**) Antapical view of hypothecal plates. Plate labels according to the Kofoidian system. cp = cover plate; Po = pore plate; vp = ventral pore; X = X-plate or canal plate. Abbreviation of sulcal plates: Sa = anterior sulcal plate; Sp = posterior sulcal plate. Scale bars = 2 μm (**A**–**H**,**K**,**L**) or 1 μm (**I**,**J**).

**Figure 4 microorganisms-09-00134-f004:**
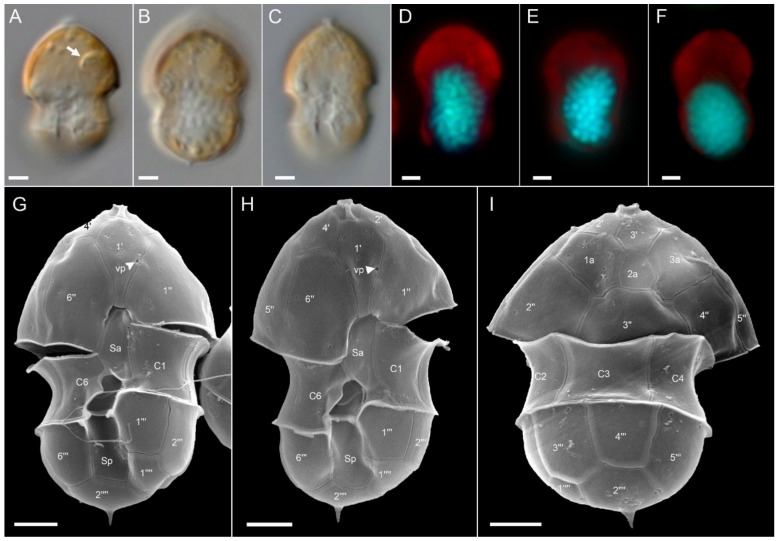
*Azadinium spinosum* ribotype B strain 5-F6. (**A**–**F**) LM images of living (**A**–**C**) cells to indicate general size and shape. Note the distinct pyrenoid in the episome (arrow in **A**). (**D**–**F**) Formalin fixed and DAPI-stained cells viewed with UV excitation to indicate shape and position of the nucleus. (**G**–**I**) SEM images of different thecae in ventral (**G**,**H**) or dorsal (**I**) view. Note position of ventral pore (vp). Plate labels according to the Kofoidian system. vp = ventral pore. Abbreviation of sulcal plates: Sa = anterior sulcal plate; Sp = posterior sulcal plate. Scale bars = 2 μm.

**Figure 5 microorganisms-09-00134-f005:**
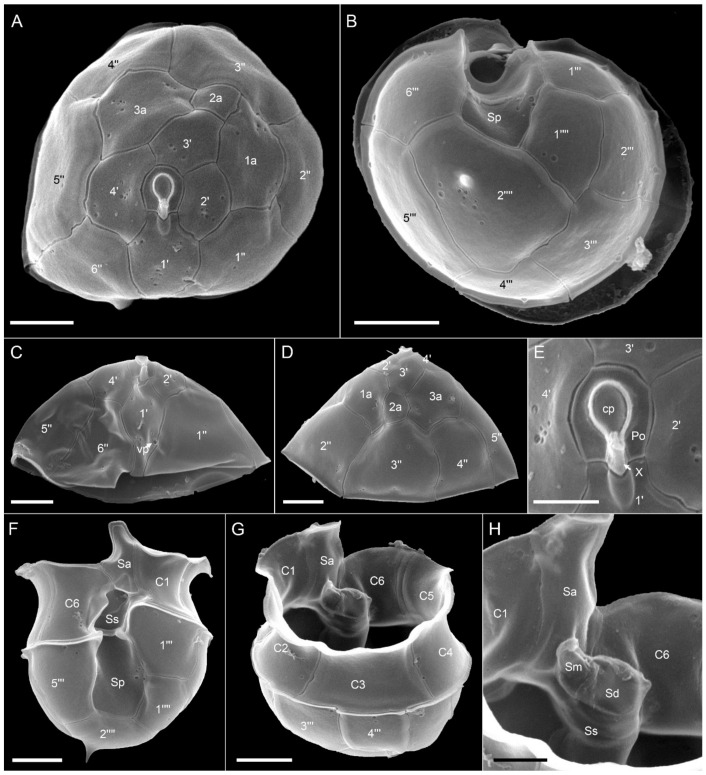
*Azadinium spinosum* ribotype B strain 5-F6. SEM images of different thecae. (**A**) Apical view of epithecal plates. (**B**) Antapical view of hypothecal plates. (**C**) Epitheca in ventral view. Note the position of the ventral pore (vp). (**D**) Epitheca in dorsal view. (**E**) Detailed view of the apical pore complex (APC). (**F**) Hypotheca in ventral view. (**G**) Hypothecal in apical/dorsal view. (**H**) Detailed internal view of the sulcal plates. Plate labels according to the Kofoidian system. cp = cover plate; Po = pore plate; vp = ventral pore; X = X-plate or canal plate. Abbreviation of sulcal plates: Sa = anterior sulcal plate; Sd = right sulcal plate; Sm = median sulcal plate; Sp = posterior sulcal plate; Ss = left sulcal plate; Scale bars = 2 μm (**A**–**D**,**F**,**G**) or 1 μm (**E**,**H**).

**Figure 6 microorganisms-09-00134-f006:**
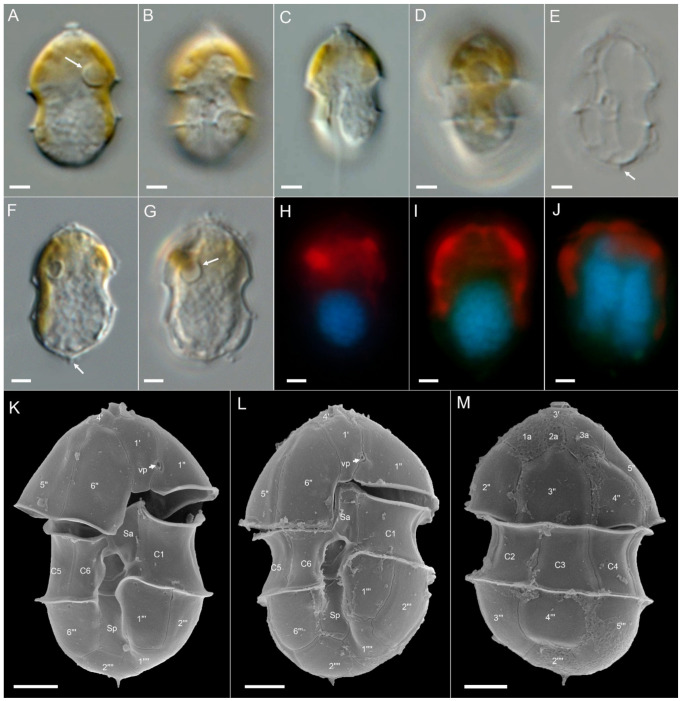
*Azadinium* cf. *spinosum* strain 5-B9. (**A**–**G**) LM images of living (**A**–**D**) or formalin fixed (**E**–**G**) cells to indicate general size and shape. Note the distinct pyrenoid in the episome (arrow in **A**,**G**) and the antapical spine (arrow in **E**,**F**). (**H**–**J**) Formalin fixed and DAPI-stained cells viewed with UV excitation to indicate shape and position of the nucleus. (**J**) Late stage of nuclear division. Note the elongated shape of the nucleus. (**K**–**M**) SEM images of different thecae in ventral (**K**,**L**) or dorsal (**M**) view. Note position of ventral po––re (vp). Plate labels according to the Kofoidian system. Abbreviation of sulcal plates: Sa = anterior sulcal plate; Sp = posterior sulcal plate. Scale bars = 2 μm.

**Figure 7 microorganisms-09-00134-f007:**
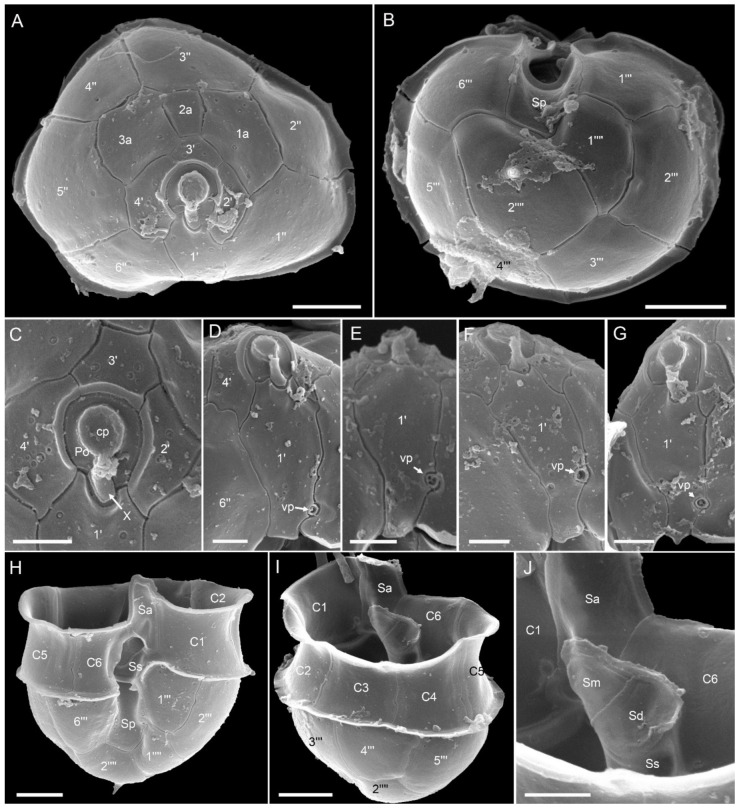
*Azadinium* cf. *spinosum* strain 5-B9. SEM images of different thecae. (**A**) Apical view of epithecal plates. (**B**) Antapical view of hypothecal plates. (**C**) Detailed view of the apical pore complex (APC). (**D**–**G**) First apical plate in ventral view. Note the position of the ventral pore (vp). (**H**) Hypotheca in ventral view. (**I**) Hypotheca in apical/dorsal view. (**J**) Detailed internal view of the sulcal plates. Plate labels according to the Kofoidian system. cp = cover plate; Po = pore plate; vp = ventral pore; X = X-plate or canal plate. Abbreviation of sulcal plates: Sa = anterior sulcal plate; Sd = right sulcal plate; Sm = median sulcal plate; Sp = posterior sulcal plate; Ss = left sulcal plate; Scale bars = 2 μm (**A**,**B**,**H**,**I**) or 1 μm (**C**–**G**,**J**).

**Figure 8 microorganisms-09-00134-f008:**
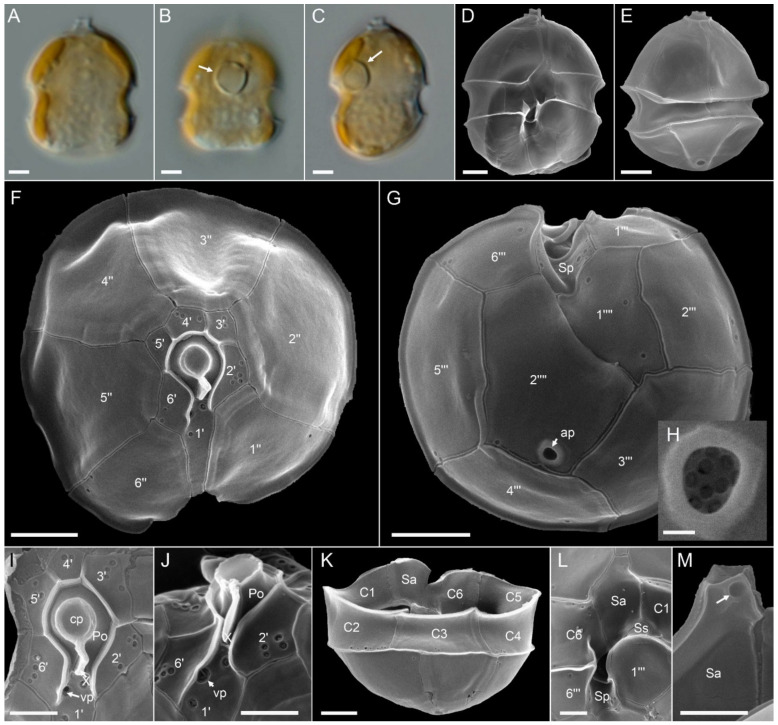
*Amphidoma languida* strains. (**A**–**C**) LM images of living cells to indicate general size and shape. Note the distinct pyrenoid in the episome (arrows in **B**,**C**). (**D**–**M**) SEM images of different thecae. (**D**) Ventral view. (**E**) Dorsal view. (**F**) Apical view of epithecal plates. (**G**) Antapical view of hypothecal plates. Note the antapical pore (ap). (**H**) Enlarged view of the antapical pore. (**I**) Detailed view of the apical pore complex (APC). (**J**) APC in ventral view, note the position of the ventral pore (vp). (**K**) Hypotheca in dorsal view. (**L**) Detailed view of sulcal plates. (**M**) Detailed view of the anterior sulcal plate. Note the anterior round ventral depression (arrow). Plate labels according to the Kofoidian system. ap = antapical pore; cp = cover plate; Po = pore plate; vp = ventral pore; X = X-plate or canal plate. Abbreviation of sulcal plates: Sa = anterior sulcal plate; Sd = right sulcal plate; Sm = median sulcal plate; Sp = posterior sulcal plate; Ss = left sulcal plate. Scale bars = 2 μm (**A**–**G**,**K**) or 1 μm (**I**,**J**,**L**,**M**) or 0.2 μm (**H**).

**Figure 9 microorganisms-09-00134-f009:**
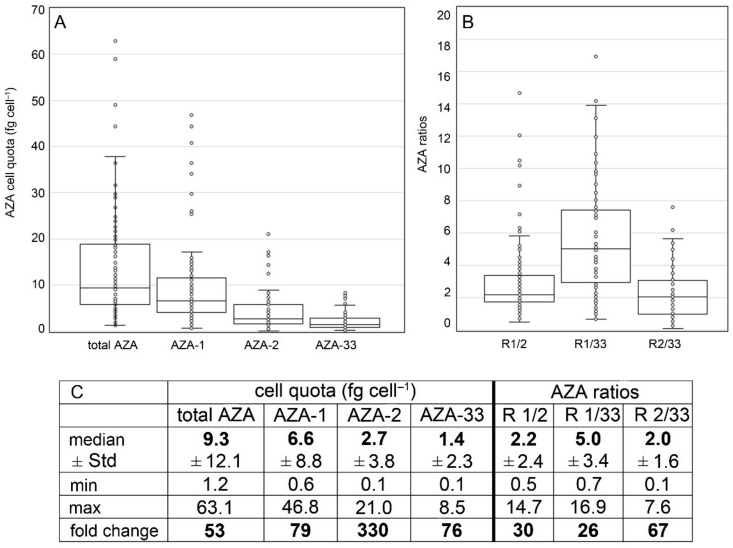
*Azadinum spinosum* ribotype A strains. Box-Whisker plots (**A**,**B**) and summary statistics (**C**) of all analyses (including repeated analyses of single strains) of AZA cell quota (**A**) and AZA ratios (**B**). Std = standard deviation.

**Figure 10 microorganisms-09-00134-f010:**
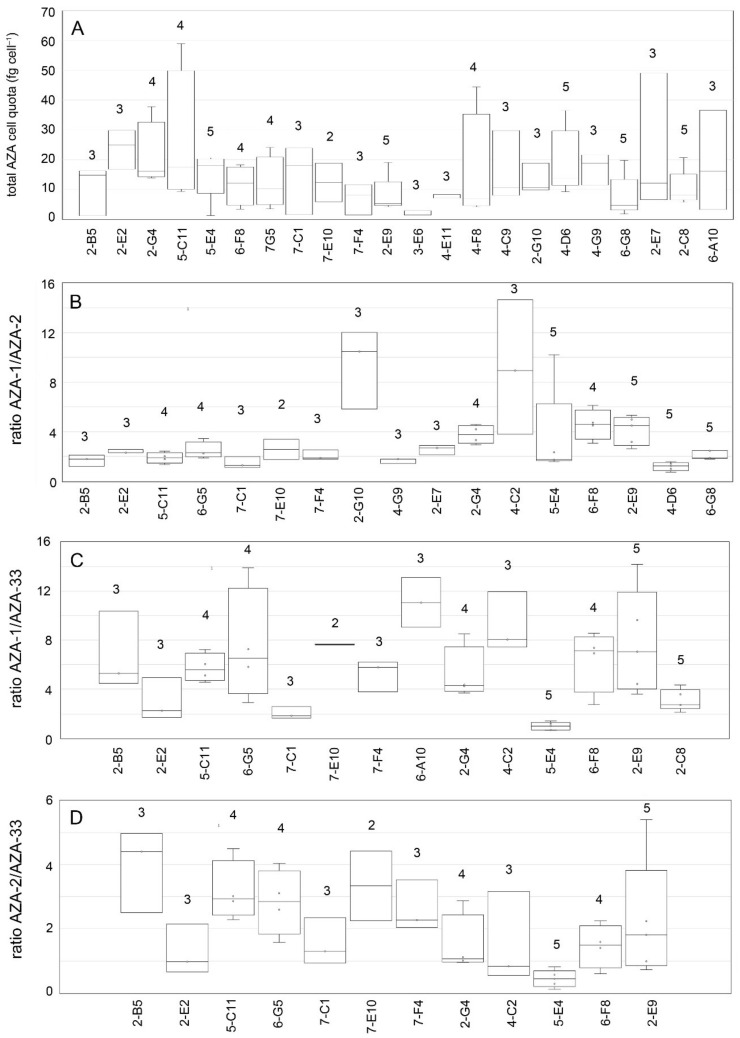
*Azadinium spinosum* ribotype A strains. Variability in total AZA cell quota (**A**) and AZA ratios (**B**–**D**) based on multiple analysis of single strains. Number above bars indicate the number of analyses (*n*).

**Figure 11 microorganisms-09-00134-f011:**
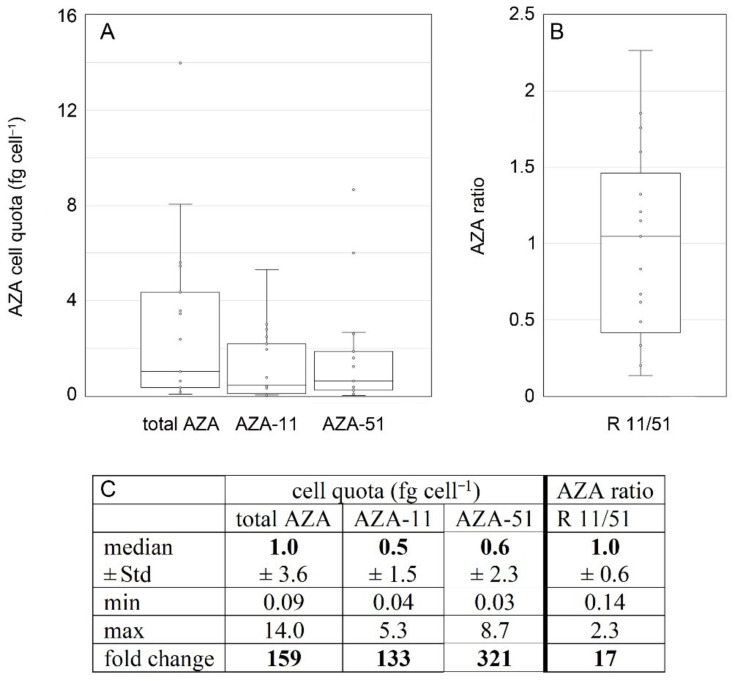
*Azadinium spinosum* ribotype B strains. Box and Whisker plots (**A**,**B**) and summary statistics (**C**) of all analyses (including repeated analyses of single strains) of AZA cell quota (**A**) and AZA ratio (**B**). Std = standard deviation.

**Figure 12 microorganisms-09-00134-f012:**
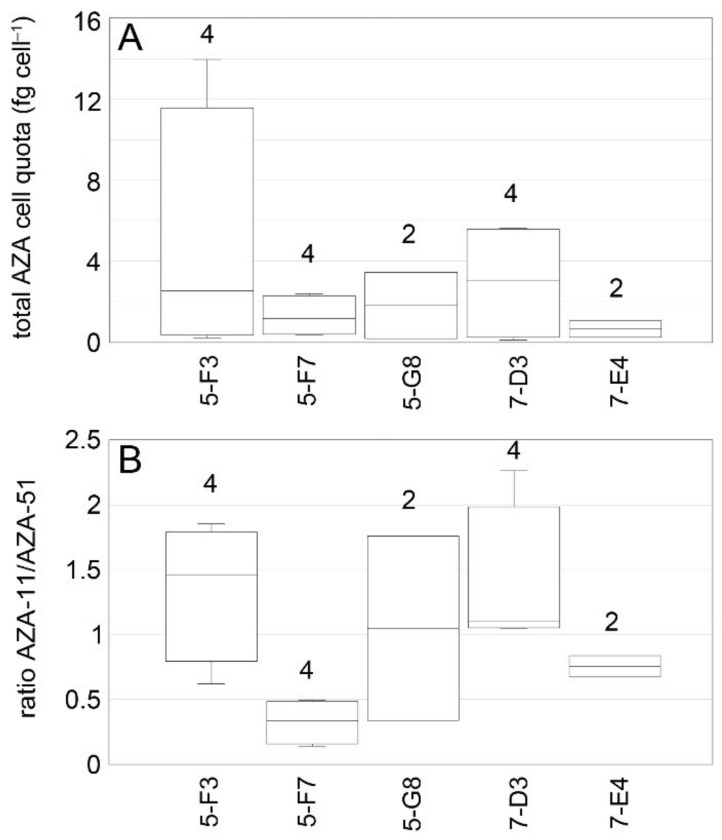
*Azadinium spinosum* ribotype B strains. Variability in total AZA cell quota (**A**) and AZA ratio (**B**) based on multiple analyses of single strains. Number above bars indicate the number of analyses (*n*).

**Figure 13 microorganisms-09-00134-f013:**
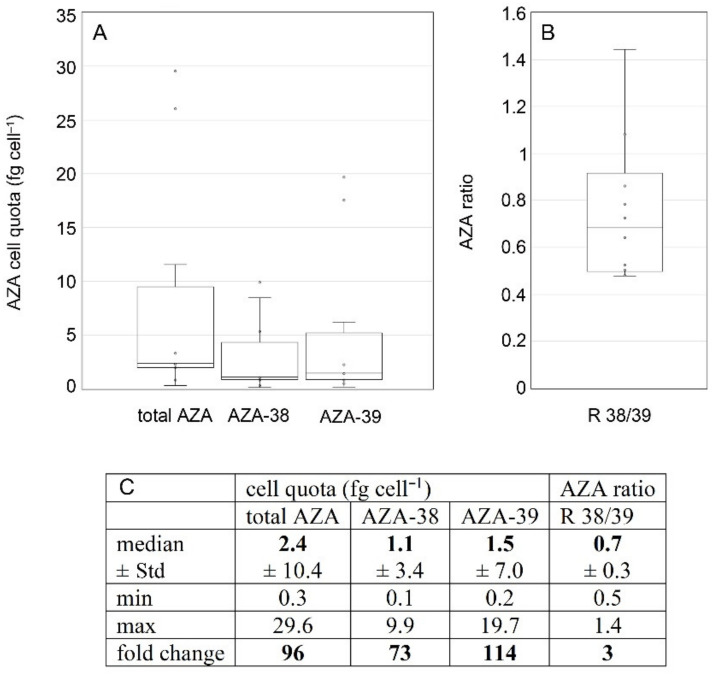
*Amphidoma languida* strains. Box-Whisker plots (**A**,**B**) and summary statistics (**C**) of all analyses of AZA cell quota (**A**) and AZA ratios (**B**). Std = standard deviation.

**Figure 14 microorganisms-09-00134-f014:**
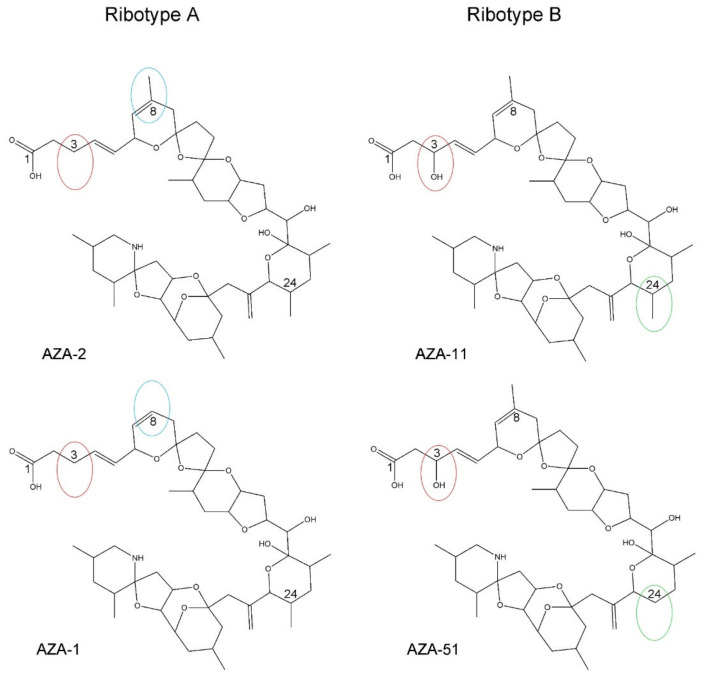
Planar structures of AZA-1, AZA-2, AZA-11, and AZA-51. All AZA are closely related and can be regarded as variants of AZA-2: AZA of ribotype B are hydroxylated at C3 (red ovals), whereas ribotype A AZA are not. AZA profiles of both ribotypes consist of a base compound (AZA-2 and AZA-11, respectively) and a demethylated variant (AZA-1 and AZA-51, respectively). The demethylation site of ribotype A AZA is C8 and of ribotype B C24 (green ovals).

**Figure 15 microorganisms-09-00134-f015:**
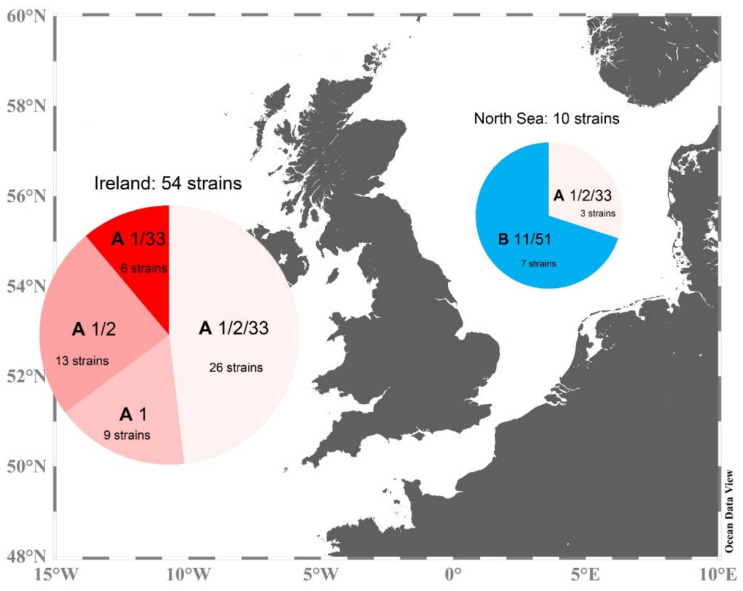
Summary of ribotype (**A**,**B**), toxin profile (1/2/33; 1/2; 1/33; 1; 11/51) and distribution of newly obtained *Az. spinosum* strains.

**Table 1 microorganisms-09-00134-t001:** Summary of Amphidomataceae strains obtained during the field sample campaign HE516 in summer 2018.

Genus	Species	No. of Strains	Reference
*Azadinium*	*caudatum* var. *margalefii*	1	Salas et al. 2021 [35]
*Azadinium*	cf. *zhuanum*	1	Salas et al. 2021 [35]
*Azadinium*	*galwayense*	3	Salas et al. 2021 [35]
*Azadinium*	*perfusorium*	26	Salas et al. 2021 [35]
*Azadinium*	*spinosum* ribotype A	60	this paper
*Azadinium*	*spinosum* ribotype B	7	this paper
*Azadinium*	cf. *spinosum*	5	this paper
*Amphidoma*	*languida*	10	this paper

**Table 2 microorganisms-09-00134-t002:** Presence and location of compensatory base change (CBC) among *Azadinium spinosum* (ribotype A, B, and C) and/or *Az.* cf. *spinosum* (cf.) strains.

Strains	7-D3 (B)	H-4-G1 (C)	1-H10 (cf.)	5-B9 (cf.)
3D9 (A)	no	no	Helix IV	Helix IV
7-D3 (B)		no	Helix IV	Helix IV
H-4-G1 (C)			Helix IV	Helix IV
1-H10 (cf.)				no

**Table 3 microorganisms-09-00134-t003:** Sequence alignment of the *Az. spinosum* specific qPCR primers and probe with the respective ribotype homologous. Base-pair differences to the primer or probe sequence are highlighted in yellow.

	F-Primer	Probe	R-Primer
Sequence	CATCTCCCTGACACAAAGACGA	AGGAGTCCTTTTGGGCG	GGAAACTCCTGAAGGG-CTTGT
ribotype A	---------------------------------------------	----------------------------------	--------------------------------------------
ribotype B	---------------------------------------------	--------------------C------------	---------------T--------G-----------------
ribotype C	---------------------------------------------	C--A----------------------------	---------------T--------TCA-------CCA
*Az.* cf. *spinosum*	---------------------------------------------	T-A----------G-----------------	T-------------T----------------A---------

**Table 4 microorganisms-09-00134-t004:** *Az. spinosum* strains, stability of toxin profile. Data display the number of strains tested for toxin profile confirmation at different time points after isolation. AZA = Azaspiracids.

	Toxin Profile Confirmation		
Toxin Profile	ca. 2 Months Later	ca. 5 Months Later	>1 Year
AZA-1, AZA-2, AZA-33	12	11	4
AZA-1	5	2	1
AZA-1, AZA-2	6	6	2
AZA-1, AZA-33	2	2	1
AZA-11, AZA-51	3	3	5
cf. *spinosum* (none)	5	4	3

**Table 5 microorganisms-09-00134-t005:** Summary statistics, analysis of variance (ANOVA) Kruskal–Wallis test. Significance level <0.05 are highlighted with grey shading.

Ribotype A	Groups: Toxin Profile (1, 2, 3, 4)		
	DF, *n*	H	*p*
total AZA	3, *n* = 125	9.81	0.0203
AZA-1	3, *n* = 125	2.16	0.5417
AZA-2	1, *n* = 94	1.05	0.3043
AZA-33	1, *n* = 76	0.51	0.4740
R 1/2	1, *n* = 94	3.68	0.0549
R 1/33	1, *n* = 76	0.03	0.8652
	**Groups: Strains**		
total AZA	21, *n* = 79	26.37	0.1927
R 1/2	16, *n* = 62	46.29	0.0001
R 1/33	13, *n* = 51	35.82	0.0006
R 2/33	11, *n* = 43	24.82	0.0097
**Ribotype B**	**Groups: Strains**		
	DF, *n*	H	*p*
total AZA	4, *n* = 16	1.09	0.8956
AZA-11	4, *n* = 16	2.91	0.5727
AZA-51	4, *n* = 16	0.97	0.9157
R 11/51	4, *n* = 16	8.60	0.0718

## Data Availability

The data presented in this study are in the article and Appendix A.

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
