# Peer review of "Multiple New Strains of Amphidomataceae (Dinophyceae) from the North Atlantic Revealed a High Toxin Profile Variability of Azadinium spinosum and a New Non-Toxigenic Az. cf. spinosum"

_microorganisms, 2021, doi:10.3390/microorganisms9010134_

Round 1
Reviewer 1 Report
The article entitled "Multiple new strains of Amphidomataceae (Dinophyceae) from the North Atlantic revealed a high toxin profile variability of Azadinium spinosum and a new non-toxigenic Az. cf. spinosum" has provided azaspiracids profiles in 82 isolates. The authors have improved the quality of the presentation from the previous version, and therefore, I recommended accepting this manuscript.
Author Response
Reviewer 1
The article entitled "Multiple new strains of Amphidomataceae (Dinophyceae) from the North Atlantic revealed a high toxin profile variability of Azadinium spinosum and a new non-toxigenic Az. cf. spinosum" has provided azaspiracids profiles in 82 isolates. The authors have improved the quality of the presentation from the previous version, and therefore, I recommended accepting this manuscript.
Reply: We very much appreciate the positive reception of our revision
Reviewer 2 Report
This manuscript describes the observations of strains of the Amphidomataceae isolated from surveys in the NE Atlantic Ocean around the British Islands. The most interesting information is the description of new species of Azadinium that the authors have submitted to another journal. This manuscript remains for the description of the morphology, molecular phylogeny and toxicity of strains of already known species. Despite the lack of originality, this information is useful because the species of Azadinium has interest for public health.
The first version had an important problem. The authors cited three species as new for science using “sp. nov.”, but they did not describe these species. This is incorrect and only creates confusion if you are describing the species simultaneously in another publication. Now, the three new species are reduced to only two new species, and they are cited in a publication –in press-. Next time, please be patient and first described the new species and later cited them in other papers.
Now the situation is clearer, but still confusing. For example, the table 1 cited 113 strains, and the map showed the 82 strains studies in this study. The reader will ask where were isolated the missing strains cited in the table 1.
Despite the over split of the results, trying to publish the results of the same survey in distinct journals, I will not oppose to the publication of these results.
Please correct the next minor aspects:
line 73: Toxigenic specimen
Please do not use the term ‘specimen’, just use cells, individuals…
Specimen in nomenclature has a more restricted meaning, as the type specimen, referring to a single cell used as type.
line 77: it is remarkable that knowledge of the diversity in Irish waters is based on two strains only, i.e. one strain of Az. spinosum [18] and one strain of Am. languida [32].
There were records of the Amphidomataceae before the description of the genus Azadinium. For example, Dodge (1982, p. 205) reported Amphidoma caudata at Galway, Ireland.
line 98: 15th August, 2018 better as 15 August 2018?
line 102: sieve, pooled, and well mixed.
I already commented that aspect. When you mix samples from distinct depths, you lost the information on the possible vertical distribution of the species.
line 118 Sanger-Sequencing of strain DNA was performed for the 18S/small subunit (SSU), D1/D2 region of 28S/large subunit
18S and 28S are unnecessary as you use SSU and LSU in the text.
line 192 ACA AGG (ACT)TC CGT AGG T-3
Please explain the meaning of the parenthesis in the primers.
line 199 Alfred-Wegener-Institute (Helmholtz Center for Polar- and Marine Research, Bremerhaven, Germany).
Please decide a single name for the Institution
line 202 Beverly, Massachusetts, USA)
line 280 StatSoft, Tulsa, USA).
If you cite the name of the state of USA, please be coherent and cited it in all the cases. It is more common to use the standard abbreviation of the state.
Figure 2. Phylogenetic tree.
Even if you reported the information as supporting material, the phylogenetic tree should include the accession numbers (also in the new sequences).
Your new sequences should include the geographic origin. For example, you cited “Norway, North Atlantic” or “Southwest Atlantic Argentina”, “Mediterranean Greece”. We know that Greece is in the Mediterranean Sea. Please cite a more precise location.
In the figure legend, if you cited the abbreviation “Bayesian inference (BI)”, please be coherent and explain the meaning of “ML”.
line 377 generally round to slightly ellipsoid but could be more elongated as well.
Confusing. You are describing an ambiguous morphology. Please explain if the elongate nucleus is exclusively due to a cell that begins the mitosis.
line 388 Po = pore plate. This is the apical pore plate.
line 390: Scale bars = 2 μm (A–H, K, L) or 1 μm (I, J).
There are no significant differences between using one and 2-μm scale bars. Please use a similar scale bar of 2 microns for all the micrographs. Please extend this to the other figures.
Figure 3 and 4. High magnification pictures of the sulcal region are missing.
Figures 3-8. Please increase the size of the type in the thecal plate labelling.
line 430: central sulcal plates Plate labels
Please check the double ‘plate’.
Central sulcal? Where are the lateral sulcals? Please remove central.
Figures 5H, 7J. Ss (left sulcal) and Sd (right sulcal) do not fit with their definitions. Sd is anterior to Ss.
Fig 7E. Are there three pores inside the ventral pore? Please provide insets with high magnification pictures of the ventral pore of distinct individuals.
Author Response
Reviewer 2
This manuscript describes the observations of strains of the Amphidomataceae isolated from surveys in the NE Atlantic Ocean around the British Islands. The most interesting information is the description of new species of Azadinium that the authors have submitted to another journal. This manuscript remains for the description of the morphology, molecular phylogeny and toxicity of strains of already known species. Despite the lack of originality, this information is useful because the species of Azadinium has interest for public health.
The first version had an important problem. The authors cited three species as new for science using “sp. nov.”, but they did not describe these species. This is incorrect and only creates confusion if you are describing the species simultaneously in another publication. Now, the three new species are reduced to only two new species, and they are cited in a publication –in press-. Next time, please be patient and first described the new species and later cited them in other papers.
Now the situation is clearer, but still confusing. For example, the table 1 cited 113 strains, and the map showed the 82 strains studies in this study. The reader will ask where were isolated the missing strains cited in the table 1.
Despite the over split of the results, trying to publish the results of the same survey in distinct journals, I will not oppose to the publication of these results.
Reply: As stated in the reply to the first version it was always the plan to have the species-description paper in press first.
In Table 1 reference to the paper where the other strains are reported is given so the reader can look there for the detailed origin of these strains.
We regret that the reviewer saw an “over-split” of results, which we cannot understand given the amount of original data presented in both papers.
Please correct the next minor aspects:
line 73: Toxigenic specimen
Please do not use the term ‘specimen’, just use cells, individuals…
Specimen in nomenclature has a more restricted meaning, as the type specimen, referring to a single cell used as type.
Reply: corrected, specimen replaced by cells
line 77: it is remarkable that knowledge of the diversity in Irish waters is based on two strains only, i.e. one strain of Az. spinosum [18] and one strain of Am. languida [32].
There were records of the Amphidomataceae before the description of the genus Azadinium. For example, Dodge (1982, p. 205) reported Amphidoma caudata at Galway, Ireland.
reply: We agree and included reference to Az. caudatum by Dodge as follows:
“…is based on field sample records of Az. caudatum [32] and two strains of Amphidomataceae only, i.e. one strain of…”
The reference list was extended accordingly.
line 98: 15th August, 2018 better as 15 August 2018?
Reply: We leave this an editorial decision depending on the journals style
line 102: sieve, pooled, and well mixed.
I already commented that aspect. When you mix samples from distinct depths, you lost the information on the possible vertical distribution of the species.
Reply: Thank you very much for this comment, and we refer to the reply made on the first round of comments.
line 118 Sanger-Sequencing of strain DNA was performed for the 18S/small subunit (SSU), D1/D2 region of 28S/large subunit 18S and 28S are unnecessary as you use SSU and LSU in the text.
Reply: 18S and 28S was removed
line 192 ACA AGG (ACT)TC CGT AGG T-3
Please explain the meaning of the parenthesis in the primers.
Reply: this (bases in parenthesis) is the common notation for degenerated primers, i.e a mix of oligonucleotide sequences in which some positions contain a number of possible bases, giving a population of primers with similar sequences that better covers possible nucleotide combinations. As this is common notation in molecular biology no explanation was added to the manuscript.
line 199 Alfred-Wegener-Institute (Helmholtz Center for Polar- and Marine Research, Bremerhaven, Germany).
Please decide a single name for the Institution
Reply: This is the official name of the Institute, Nevertheless, we here removed the “Helmoltz Center for Polar- and Marine Research”
line 202 Beverly, Massachusetts, USA)
line 280 StatSoft, Tulsa, USA).
If you cite the name of the state of USA, please be coherent and cited it in all the cases. It is more common to use the standard abbreviation of the state.
Reply: Standard abbreviations for the US state are now consistently used.
Figure 2. Phylogenetic tree.
Even if you reported the information as supporting material, the phylogenetic tree should include the accession numbers (also in the new sequences).
Your new sequences should include the geographic origin. For example, you cited “Norway, North Atlantic” or “Southwest Atlantic Argentina”, “Mediterranean Greece”. We know that Greece is in the Mediterranean Sea. Please cite a more precise location.
Reply: Please see our reply to the first round of comments. Accession numbers are given in the supplement, and if there is no format requirement of the Journal Mikroorganisms for a mandatory inclusion of accession numbers in phylogenetic trees we leave it as it is.
The origin of strains is given as (Oceanographic area, country), so there is nothing wrong with e.g. Mediterranean, Greece. Nothing was changed.
In the figure legend, if you cited the abbreviation “Bayesian inference (BI)”, please be coherent and explain the meaning of “ML”.
Reply: We added maximum likelihood (ML).
line 377 generally round to slightly ellipsoid but could be more elongated as well.
Confusing. You are describing an ambiguous morphology. Please explain if the elongate nucleus is exclusively due to a cell that begins the mitosis.
Reply: Changed to:
“…but could be, presumably in early stages of cell division, more elongated…”
line 388 Po = pore plate. This is the apical pore plate.
Reply: We follow the definitions given in Fensome et al. 1993, page 250: ”….two plates: a pore plate (PO, or peripherial apical platelet), surrounding…”
Nothing was changed.
line 390: Scale bars = 2 μm (A–H, K, L) or 1 μm (I, J).
There are no significant differences between using one and 2-μm scale bars. Please use a similar scale bar of 2 microns for all the micrographs. Please extend this to the other figures.
Reply: We do not agree. Extending the scale of J to 2 µm would result in a too wide scale bar, whereas reducing the scale in e.g. A is no option as well. Nothing was changed.
Figure 3 and 4. High magnification pictures of the sulcal region are missing.
Reply: For sulcal plates of Az. spinosum ribotype A we refer to the original papers where the details are described and depicted. Details of the sulcus for ribotype B strains (”missing in Fig. 4” are given in Fig. 5 F–H. Nothing was changed.
Figures 3-8. Please increase the size of the type in the thecal plate labelling.
Reply: Plate label font size was increased and new figures 3, 5, 7, 8 were inserted.
line 430: central sulcal plates Plate labels
Please check the double ‘plate’.
Central sulcal? Where are the lateral sulcals? Please remove central.
Reply: Corrected, “central” was removed.
Figures 5H, 7J. Ss (left sulcal) and Sd (right sulcal) do not fit with their definitions. Sd is anterior to Ss.
Reply: We agree that the orientation and denomination of sulcal plates is open for discussion and that there might be different ways how authors interpret different plates. But there is no “right or wrong” here and we keep the plate label denomination as they are.
Fig 7E. Are there three pores inside the ventral pore? Please provide insets with high magnification pictures of the ventral pore of distinct individuals
Reply: No, there are no “pores” inside the ventral pore of Amphidomataceae, in Fig, 7E is just some debris and/or remains of membraneous material as is also present on the plate surface. Providing more high-magnification micrographs of the ventral pore is outside the scope of this paper and would unnecessary increase the number of figure plates. Nothing was changed here.
This manuscript is a resubmission of an earlier submission. The following is a list of the peer review reports and author responses from that submission.
Round 1
Reviewer 1 Report
"Multiple new strains of Amphidomataceae (Dinophyceae) from the North Atlantic revealed a high toxin profile variability of Azadinium spinosum and a new non-toxigenic Az. cf. spinosum" identified 82 new AZA producing strains from waters around Irish and British areas. I felt the information provided is sufficient as well as the methods were sound. A couple points I have mainly focus on the numbers of figures/tables and the Discussion section. I suggest at least figures (10,12,14) and tables (2,3) can be moved to the supplemental materials, and the number of dinoflagellate plates can also be reduced. Line 300 24 Az. spinosum or 72? The table had 72 Az. spinosum strains in total. Line 556 Is this a misplaced subtitle? Line 557-569 I do not understand why were authors comparing the differences of AZA structures? No compound identified in this paper was unique to those have been published. Line 586-594 I believe what authors wanted to state was "in most cases, ribotype A contained higher cell quota of AZA than ribotype B, indicating ribotype A was more toxic than ribotype B." However, the statement as now is too complicated and confusing. Figure 9 the summary stat (C) is not a figure. Figure 10 can be moved to supplemental materials. Figure 11 the summary stat (C) is not a figure. Figure 12 can be moved to supplemental materials. Figure 13 the summary stat (C) is not a figure. Figure 14 can be moved to supplemental materials.
Reviewer 2 Report
This manuscript describes the morphology and toxicity of strains of the genus Azadinium isolated from the North Sea and waters around Ireland. This is the region where the genus was first described and there are numerous previous studies. The originality is low. In the last 10 years, there is a saturation of papers describing new species of the Amphidomataceae based on subtle genetic differences using high variable markers (ITS). The existence of varieties is accepted for Azadinium caudatum while ignored for other species of Azadinium?
This manuscript has the problem of the overlapping with the contents with other unpublished manuscript:
Salas, R.; Tillmann, U.; Gu, H.; Wietkamp, S.; Krock, B.; Clarke, D. Morphological and molecular characterization of three new Azadinium species revealed a high diversity of non-toxigenic species of Amphidomataceae in Irish waters, North East Atlantic. Phycol. Res.
The report of A. pseudozhuanum sp. nov., A. galwayense sp. nov. and A. perfusorium sp. nov. in the Table 1 is incorrect. The use of “sp. nov.” means that the authors are describing new species, but the formal descriptions are missing here. Even, in the phylogenetic tree of the figure 2, these undescribed species are cited as correct names, but these species do not exist yet. This is an example of the oversplit of the results and the impatience of the authors. If the species have not been formally described, these species do not exit and you must refer to them as “Azadinium sp. ined. “. The correct protocol is that the authors will wait to the publication of the other article because we do not know if the descriptions of A. pseudozhuanum, A. galwayense and A. perfusorium will be accepted and the final name for these species. There are no accessions numbers for the DNA sequences associated with these species in the tree. Please include the accession numbers in the tree (even if present in a table in the supplementary material).
The usual procedure is to report the phylogenetic tree with a single molecular marker using distinct trees for the SSU, LSU and ITS rRNA gene sequences. This allows to compare the coherence of the topology in both the SSU- and LSU rRNA gene phylogenies. This is an electronic journal, there are no space limitations.
Obviously, the use of the hypervariable ITS marker resulted in distinct clades, which represents populations more than species. If you use varieties for A. caudatum, what not for the other species?
The co-existence of multiple species with identical morphological and ecological requirements in a same location seems to be contrary to the principle of competitive exclusion. What do you mix the samples from distinct depths? Each species may have a preferential depth.
The table 4 shows that the toxicity of the strains disappear with the time. It is evident that the presence of toxins is not stable, and its production is induced by the environmental conditions. If a strain is toxic a few days after collection, and the toxicity disappears after several weeks, it has no sense to continue with the concept of toxic and to focus on a more stable characters (as the genes able to produce toxins). The strains were isolated from natural waters. The correct protocol is to report the toxicity found in the waters where the strains were isolated. That information is missing or probably the authors will publish these data in other paper.
Taken into account that these species received attention because they can affect to the industry of the mussels or oysters in the shoreline. Numerous samples were collected in the open sea, more than 100 Km far from the places where the toxicity cannot affect to public health. The efforts should be focus on the shoreline where the potential problems are.
My general comments is that authors should avoid the oversplit of the results. As the authors have submitted their results to Phycological Research, they should wait to publish these results. This reviewer does not know what results are already published in the other journal. You cannot cite the new species “nov. sp.” if you are not describing here the new species. You cannot include species names that do not exist yet in the phylogenetic tree.
Minor comments:
line 80: evaluate the potential of local non-toxigenic species/strains for false positive signals either in LM based and/or PCR based monitoring programs.
Is there any phytoplankton observer able to identify the Azadinium species based on LM routine observations? If yes, why the widespread and diverse genus Azadinium was described only ten years ago?
Line 99: At each station, plankton samples were collected with 10 L Niskin bottles at 3 m, 10 m and the depth-chlorophyll-maximum 100 (DCM) layer. Five liters of seawater from each depth were filtered through a 20 μm mesh-size Nitex 101 sieve, pooled, and well mixed.
Why do you collect the samples at distinct depth and later you mix the samples?
You lost the information of the depth where each species live. You are losing eco-physiological information.
Line 119: 0.2 mL filtered seawater…Isolation plates from the cruise were inspected after two weeks.
Do you mean that the cells grow in seawater without additional nutrients during two weeks?
line 125: Antarctic seawater
Is logical to transport seawater from the Antarctic Ocean for growing strains of the North Atlantic? Greta Thunberg will be upset with people that produce CO2 transporting seawater for long distances. Your strains from the North Atlantic Sea do not like the North Atlantic water?
line 170: filters (Millipore, 25 mm Ø, 3 mm pore-size)
Please check 3 mm.
Line 190 1F (5′ − AAC CTG GTT GAT CCT GCC AGT − 3′) and 1528R (5′
The primer 1528R provides the full SSU rDNA sequence? what the differences with Medlin et al. 1988’s primers (EukA-EukB) are?
line 204. a Nexus Gradient Mastercycler (Eppendorf) with conditions described in Tillmann et al. [37].
line 208 described in Tillmann et al. [38].
Taking into account that this is an electronic journal with less space limitation, you can add a few lines with the description of the PCR conditions.
line 275: Irish coast and from the central North Sea
Why from the central North Sea that is far from the coast? The toxicity affects to the mussels and oysters that are collected in the shoreline.
line 287 table 1
pseudozhuanum sp. nov. etc.
If you report “sp. nov.”, you must describe the new species here.
Figure 1. The scale of bathymetry is irrelevant.
line 304: 11bp split
Figure 2. Phylogenetic tree.
As reported above, in addition to this tree, please report the SSU, LSU and ITS gene sequences in independent trees. If you only include sequence of the Thoracosphaeraceae, you are forcing a phylogenetic relationship with Azadinium. Please include more dinoflagellates of other groups in SSU- and LSU rRNA trees.
The list of abbreviations in the figure legends should be in alphabetic order: vp = ventral pore; ap = antapical pore.
line 722: U.T., S.W., B.K., H.G.; resources,
Samples were collected and analyses were done in Europe. Just a curiosity, what is the contribution of the Chinese co-author?